# Rethinking Data Curation in LLM Training: Online Reweighting Offers Better Generalization than Offline Methods

**Wanru Zhao[1]\*, Yihong Chen[2], Yuzhi Tang[3]†, Wentao Ma[3]†, Shengchao Hu[4],
Shell Xu Hu[5], Alex Iacob[1], Abhinav Mehrotra[5], Nicholas D. Lane[1]**

[1] University of Cambridge,    [2] OATML, University of Oxford,
[3] University of Toronto,    [4] Shanghai Jiao Tong University,    [5] Samsung AI Center

## Abstract

Data curation is a critical yet underexplored area in large language model (LLM) training. Existing methods, such as data selection and mixing, operate in an *offline* paradigm, detaching themselves from training. This separation introduces engineering overhead and makes the curation brittle: the entire pipeline must be re-run under model/task shifts. Moreover, offline methods alter data size through hard filtering or resampling, often sacrificing data diversity and harming generalization.

We propose to rethink data curation as an *online* reweighting problem, where sample importance is dynamically adjusted during training via loss weighting rather than static pre-processing. Specifically, we introduce **ADAPT** (**A**daptive **D**ata reweighting for **P**retraining and Fine**T**uning), a dynamic online framework that reweights training samples with adaptive per-sample learning rates guided by similarity-based quality signals, without changing the number of training samples. Unlike offline methods that enforce a static data distribution, ADAPT acts as an implicit curriculum learner, progressively shifting focus from coarse-grained patterns to fine-grained semantic distinctions as the model evolves. Experiments on both instruction tuning and large-scale pretraining show that ADAPT consistently outperforms offline selection/mixing and prior online methods, achieving stronger cross-benchmark generalization under equal FLOPs.

🤗 https://huggingface.co/DATA-ADAPT
 https://github.com/Ryan0v0/ADAPT

## 1 Introduction

Large language models (LLMs) have demonstrated remarkable capabilities across diverse tasks (Brown et al., 2020; Chowdhery et al., 2022; Touvron et al., 2023a), yet their generalization usually hinges on the quality of the training data, including its diversity and mixing (Hoffmann et al., 2022; Kaplan et al., 2020a). While some recent efforts seek to circumvent the data quality issue and improve generalization through architectural interventions (Peyrard et al., 2022; Chen et al., 2023c; Bengio et al., 2025), most pragmatic practices still rely on curating the "right" dataset for the model and the task. Data curation – deciding what data to keep and how to weight it – has thus become a cornerstone of large language model development. On the other hand, the process of data curation has largely remained a secret sauce with pipelines relying on opaque heuristics, ad-hoc engineering choices, or costly trial-and-error, rather than principled frameworks.

Current approaches to data curation largely follow an *offline* paradigm, falling into two camps: data selection, which keeps a subset of valuable examples, and data mixing, which adjusts sampling frequencies to rebalance distributions. Despite their differences, both approaches typically follow a multi-stage pipeline: (1) train or extract features with a proxy model, (2) compute quality signals as selection or weighting criteria on a validation set, and (3) retrain the main model on the curated data. While effective in some cases, this paradigm suffers from fundamental drawbacks (Sec. 4) due to its static and subtractive nature. First, it ignores training dynamics: the value of a data sample is not

---

\*Corresponding to: Wanru Zhao (wz341@cam.ac.uk)
† Equal contribution.

static but changes as the model learns. Offline selection freezes this value based on a proxy model, creating a misalignment with the training model's evolving needs. Second, it compromises diversity: by relying on hard filtering to select a fixed subset, offline methods risk discarding the broad data distribution essential for robust generalization. Consequently, the resulting subsets often overfit to specific benchmarks and are brittle, which requires the entire pipeline to be re-run whenever the model architecture or corpus changes.

In contrast, we propose a paradigm shift to *online reweighting*, which fundamentally differs from offline selection not just in *when* the signal is computed, but also in *how* the data is used. Instead of hard-filtering a subset based on static scores, online reweighting retains the full dataset to preserve diversity, but dynamically modulates each sample's contribution via loss weighting. This preserves data diversity compared to offline's "hard cuts" while intelligently controlling each sample's "step size" in parameter updates to adapt to evolving model states.

To solve the limitation of offline data curation, we propose a unified online reweighting framework, **ADAPT** (**A**daptive **D**ata reweighting for **P**retraining and Fine**T**uning), which aligns training samples with a validation set via similarity-based weighting. Our method incurs nearly zero additional overhead yet achieves superior cross-benchmark generalization. Under a unified evaluation protocol, it consistently outperforms both offline selection/mixing strategies and prior online reweighting methods, across large-scale pretraining and instruction tuning setups.

**Our contributions.**    This work makes the following contributions:

1. 💡 **Formalization and unification of dataset curation pipelines.**    We present a unified view of data curation, showing that data selection, mixing, and balancing can all be expressed as special cases of *data reweighting*. This perspective shifts the focus from offline pre-processing to dynamic, in-training adjustment of data weights. Which offers a fair protocol we propose to view and evaluate them through a unified lens of *online data reweighting*.

2. ✒️ **Online sample-level reweighting method.**    We introduce a new online global reweighting algorithm that is effective under this unified formalization and protocol. It employs an adaptive per-sample learning rate guided by sample quality signals, without explicitly altering the dataset size, while dynamically adjusting sample weights during training.

3. 📊 **Better performance and generalization across tasks.** On both instruction tuning and large-scale pretraining experiments, our framework achieves up to up to 7.2% higher accuracy per FLOP than offline baselines including LESS (Xia et al., 2024), DoReMi (Xie et al., 2023b), and Reg-Mix (Liu et al., 2024), and other online reweighting method (Sow et al., 2025), and demonstrates 6.1% stronger out-of-domain generalization across benchmarks.

## 2    BACKGROUND AND RELATED WORK

### 2.1    DATA CURATION FOR LARGE LANGUAGE MODELS

Researchers and engineers usually rely on web crawls to gather large datasets for training large language models (Brown, 2020; Computer, 2023; Penedo et al., 2024; Tang et al., 2024; Bai et al., 2023; Kandpal et al., 2025). However, these crawls often include a substantial amount of low-quality or irrelevant content, which makes data curation necessary to build high-quality training sets. Most data curation efforts focus on methods for improving model performance (Raffel et al., 2020; Brown, 2020; Rae et al., 2021; Penedo et al., 2023; Soldaini et al., 2024), including filtering by language (Raffel et al., 2020; Xue et al., 2020), heuristic-based filtering (Gao et al., 2020; Rae et al., 2021; Penedo et al., 2023; Soldaini et al., 2024), quality filtering (Du et al., 2022; Xie et al., 2023c), data deduplication (Lee et al., 2021) and mixing (Xie et al., 2023a; Soboleva et al., 2023; Albalak et al., 2023). However, current approaches to curating such datasets are generally ad-hoc. We aim to develop a principled and automated method for data curation that can also unify different processing stages.

**Pretraining data curation.**    Several recent studies (Xie et al., 2023a; Chen et al., 2023b; Fan et al., 2023; Thakkar et al., 2023) have explored various reweighting techniques to enhance the generalization and efficiency of language models pretraining. For instance, Xie et al. (2023a) and Fan et al. (2023) are aiming at finding the optimal mixture of pretraining corpora to enhance performance

across domains. Chen et al. (2023b) propose an ordered skill learning method for data selection measuring how effectively it teaches interdependent skills for continual pretraining and fine-tuning. Although effective, these works are aiming at the group level, whereas our work explores reweighting at the sample level, offering fine-grained control during model training dynamics. Instance-level reweighting has been used in pretraining settings of LLMs (Chen et al., 2024; Jiang et al., 2024), where each sample per mini-batch is weighted over how individual samples are treated based on their loss values. In contrast, our work studies the effects of various adaptive learning rate with different quality signals considering the model states to enhance both performance and generalization for LLMs pretraining and fine-tuning regimes.

**Instruction data curation.** Research has demonstrated that prioritizing data quality and diversity over quantity is more helpful for instruction-following capabilities (Cao et al., 2023; Chen et al., 2023a; Bukharin & Zhao, 2023; Du et al., 2023; Liu et al., 2023; Li et al., 2023). Instruction tuning data includes task-based datasets curated from traditional NLP tasks (Wang et al., 2022; Sanh et al., 2022; Wei et al., 2022a; Longpre et al., 2023), and open-ended datasets (Taori et al., 2023; Conover et al., 2023; Köpf et al., 2023; Xu et al., 2023; Mukherjee et al., 2023; Zhou et al., 2023; Ding et al., 2023). In our work, we are focusing on adaptively adjust the data weight during instruction tuning without explicitly reduce the number of instruction data samples and obtain model with generalization.

## 2.2 COMMONLY USED QUALITY SIGNAL FOR DATA CURATION

- **Lexical Similarity (BM25)** (Silva & Barbosa, 2024) quantifies term-based overlap between training and validation data through sparse retrieval scoring:

$$s_{\text{BM25}}(x) = \frac{1}{|\mathcal{D}_{\text{val}}|} \sum_{v \in \mathcal{D}_{\text{val}}} \text{BM25}(x, v). \tag{1}$$

This metric captures surface-level textual similarity without semantic understanding.

- **Semantic Similarity (Embedding)** (Rubin et al., 2021) measures dense representation alignment using pretrained encoders:

$$s_{\text{Embed}}(x) = \frac{1}{|\mathcal{D}_{\text{val}}|} \sum_{v \in \mathcal{D}_{\text{val}}} \cos\big(\phi(x), \phi(v)\big), \tag{2}$$

where $\phi(\cdot)$ denotes a frozen embedding model. This approach captures semantic proximity beyond lexical overlap.

- **Distributional Alignment (Perplexity)** (Antonello et al., 2020) evaluates likelihood under a reference language model $\theta_0$:

$$s_{\text{PPL}}(x) = -\log P_{\theta_0}(x). \tag{3}$$

Lower perplexity indicates stronger distributional alignment with the reference corpus.

- **Gradient-Based Influence** (Xia et al., 2024) estimates training utility through first-order approximation of validation loss reduction:

$$s_{\text{Grad}}(x) = \big\langle \nabla_\theta \ell(x; \theta_0), \nabla_\theta \ell(\mathcal{D}_{\text{val}}; \theta_0) \big\rangle, \tag{4}$$

where $\theta_0$ denotes a proxy model. This metric directly quantifies how training on $x$ influences performance on the validation set $\mathcal{D}_{\text{val}}$.

## 3 UNIFIED FORMALIZATION OF DATA CURATION

**Setup.** Given two datasets: the *train* set $\mathcal{D}_{\text{train}}$ and the *validation* set $\mathcal{D}_{\text{val}}$. Usually, the size of the validation set is much smaller than the train set. When $\mathcal{D}_{\text{val}}$ is sampled from the distribution of downstream test data, the validation dataset $\mathcal{D}_{\text{val}}$ is considered as *in-domain*. The train dataset $\mathcal{D}_{\text{train}}$ on the other hand consists of both in-domain and *out-of-domain* samples. This is the case when web crawling is used to collect training data from the whole internet. Our goal is to subsample or weight the train set $\mathcal{D}_{\text{train}}$ under the guidance of the validation set $\mathcal{D}_{\text{val}}$, so that the model training is less affected by the out-of-domain samples.

## 3.1 Design Space of Data Quality Signal

In Sec. 2, we present a unified framework for data quality assessment where *scoring function* takes the form $s(x) \equiv s(x; \theta, \mathcal{D}_{\text{val}})$ that assigns a quality signal for each data example $x$, which optionally depends on a (proxy) model $\theta$ and the validation set $\mathcal{D}_{\text{val}}$.

## 3.2 Offline Data Curation

The scores are employed once before training (e.g., with a proxy model $\theta_0$ and fixed validation set $\mathcal{D}_{\text{val}}$). The resulting weights $\{w(x)\}_{x \in \mathcal{D}_{\text{train}}}$ are fixed and used to make a decision to pass which training examples to the real training stage.

- **Data Selection (sample-level binary weights)** Data selection (or filtering) removes part of the training corpus before pretraining begins. It keeps data examples with quality signal above a certain threshold $\tau$:
$$\mathcal{D}'_{\text{train}} = \{x \in \mathcal{D}_{\text{train}} \mid s(x) \geq \tau\}.$$

  From a data reweighting perspective, filtering data points is equivalent to assigning binary weights of the form $w(x) = \mathbf{1}[s(x) \geq \tau] \in \{0, 1\}$.

- **Data Mixing (domain-level fractional weights)** Data mixing can be seen as a coarse-grained data reweighting method that operates at the domain level. In other words, data points within the same domain receive the same quality score. Pulling all domain scores $\{s_d\}_{d \in \text{domains}}$ and normalize them to obtain data mixing probability for each domain $d$
$$w_d = \frac{g(s_d)}{\sum_{d'} g(s_{d'})},$$

  where $g$ transform the score to be non-negative with a common choice being $g : s \mapsto \exp(s)$. As such, $w_d$ determine how much of a total training budget $B$ is allocated to each domain: $B_d = w_d B$. Alternatively, training can be implemented by sampling domains with probability $w_d$ (probability mixing) or by assigning a fixed quota $B_d$ (quota mixing). Since $w_d$ is decided before training, mixing is an *offline* operation that alters the effective number of examples each domain contributes.

## 3.3 Online Data Curation

In this setting, the scoring function depends on the evolving model state $\theta_t$ and, in some cases, on a dynamic validation set. The resulting weights $\{w_t(x)\}_{x \in \mathcal{D}_{\text{train}}}$ evolve throughout training, adjusting the *gradient contributions* of examples while preserving the full volume of the training set $\mathcal{D}_{\text{train}}$.

**Data Reweighting (sample-level fractional weights)** Data reweighting assigns normalized weights to examples while keeping the training set size unchanged. Similar to Sec. 3.2, the quality score for example $x$ is transformed to a non-negative weight: $w(x) = g(s(x))$, which scales their contribution to the loss:
$$\mathcal{L}^*(\theta) = \frac{1}{Z} \sum_{x \in \mathcal{D}} w(x)\, \mathcal{L}(\theta; x), \qquad Z = \sum_{x \in \mathcal{D}} w(x).$$

Equivalently, from a stochastic gradient descent perspective, the stochastic gradient w.r.t. $x$ is scaled by $\frac{w(x)}{Z}$, which plays a role of *per-sample* learning rate in addition to the *global* learning rate $\eta$:
$$\theta_{t+1} = \theta_t - \eta \frac{w(x)}{Z} \nabla_\theta \ell(\theta_t; x).$$

Since the total number of training examples remains unchanged, only their relative contribution to parameter updates is modified, data reweighting is inherently an *online* method.

## 3.4 Unified Evaluation of Total FLOPs (Offline vs. Online)

The unification enables us to compare different methods under the same FLOPS calculation framework. We evaluate curation methods under a *cost-aware* metric that accounts for all computation spent to obtain and use the curated data. Let $B$ denote a training budget (tokens or update steps).

**Total FLOPs for offline data curation** Offline curation modifies the *amount of data that participates in training* before training starts. Its total compute is

$$F_{\text{total}}^{\text{off}} = \underbrace{F_{\text{prep}}^{\text{off}}(\mathcal{D}; f)}_{\text{data scoring as preprocessing}} + \underbrace{F_{\text{train}}(\mathcal{D}', B')}_{\text{model training on the sampled subset}}, \qquad (5)$$

where (i) $F_{\text{prep}}^{\text{off}}$ includes any corpus-wide scoring, proxy-model passes, retrieval or filtering necessary to construct $\mathcal{D}'$; (ii) $\mathcal{D}' \subseteq \mathcal{D}$ is the retained subset (for selection), or an *effective* subset induced for data mixing; (iii) $B'$ is the effective training budget after curation. By construction, offline selection uses *binary* example weights (keep/drop), and offline domain mixing changes per-domain sampling rates, thereby reducing or reallocating the volume of data seen during training.

**Total FLOPs for online data curation** Online curation *does not change the amount of training data*; instead, it modulates each example's contribution *during training*. Its total compute is

$$F_{\text{total}}^{\text{on}} = \underbrace{F_{\text{train}}(\mathcal{D}, B)}_{\text{model training on the full corpus}} + \underbrace{F_{\text{metrics}}^{\text{on}}}_{\text{on-the-fly features}}, \qquad (6)$$

where $F_{\text{metrics}}^{\text{on}}$ denotes lightweight, in-training computations needed to obtain scores (e.g., using current logits, per-example loss, or gradient norms). Because scoring is amortized inside the training loop and no data are removed, $F_{\text{metrics}}^{\text{on}}$ is typically modest relative to $F_{\text{train}}$.

Conversely, online reweighting maintains the full training signal while adding only minimal in-loop metrics computation, providing a clearer accuracy–compute trade-off when measured by *total* FLOPs.

## 4    REVISITING OF OFFLINE DATA SELECTION

In this section, based on the evaluation protocol derived from the unification above, we revisit the accuracy–FLOPs trade-off of existing offline data selection methods.

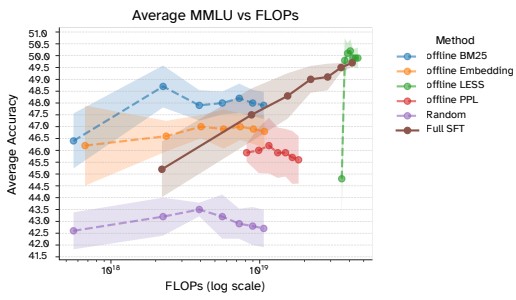
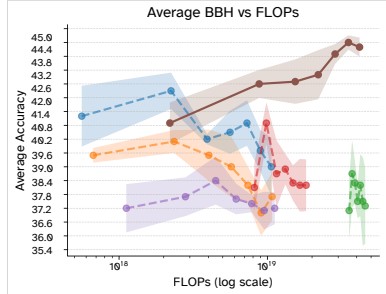

*(a)* In-domain evaluation: MMLU validation set → MMLU benchmark.

*(b)* Out-of-domain evaluation: MMLU validation set → BBH benchmark.

*Figure 1*. Efficiency–accuracy tradeoff of different data selection methods and full finetune method under our proposed **total FLOPs** metric. Selection methods (e.g., `LESS`) appear competitive under in-domain evaluation, but their advantage vanishes or even reverses in out-of-domain settings. In contrast, (`full sft`) remains consistently strong across domains.

**Observation.** Figure 1 illustrates the limitation of offline data selection methods: **Overfitting to the validation benchmark.** Model trained on offline selected data with MMLU (Hendrycks et al., 2021) as validation set performs well on the MMLU benchmark (Figure 1a), but the same model generalize poorly to BBH benchmark (Suzgun et al., 2022) (Figure 1b). This issue is especially severe when using LESS (Xia et al., 2024). This reveals that offline methods often *overfit to the chosen validation task*, lacking a true generalization. However, we observed that vanilla full dataset training demonstrate more stable performance in both benchmark. One possible reason is that official data curation which we directly change the number of data involved in the training by repetition would cause the model to replace generalization ability with memorization (Hernandez et al., 2022).

In summary, the offline paradigm has severe limitations, being neither cost-effective nor generalizable. This motivates us to explore an online alternative, which may alleviate memorization while retaining more data to enhance generalization. In the next section, we leverage these insights to design an *online reweighting method*.

## 5 Adaptive Learning Rate for Online Data Reweighting

The limitations of offline data selection suggest that a different paradigm is needed: one that is generalizable, and adaptive to the evolving state of the model. To this end, we propose an **online data reweighting framework** that is also suitable for data selection, data mixing with a unified formalization. Unlike offline methods that commit to a fixed subset before training begins, our framework dynamically adjusts data weights as the model learns, so it naturally adapts to new model states without re-running the data preprocessing and training pipeline from scratch.

### 5.1 Per-Sample Learning Rate Update

We cast data selection and mixing as *online data reweighting*. At training step $t$, given a minibatch $B_t \subset \mathcal{D}$, we update model parameters as

$$\theta_{t+1} = \theta_t - \eta \sum_{i \in B_t} w_t(i) \, \nabla_\theta \ell\big(f_\theta(x_i), y_i\big), \tag{7}$$

where $w_t(i) \geq 0$ denotes a dynamic weight assigned to sample $i$. Here, $w_t(i)$ directly scales the gradient contribution of sample $i$, and thus acts as a *per-sample learning rate multiplier*: larger weights amplify the effective step size on informative examples, while smaller weights downweight less useful ones.

In our framework, weights are derived from scoring functions $s_t(v, i)$ that compare a training example $i$ with validation/query points $v \in \mathcal{D}_{\text{val}}$. These scores are aggregated across $v$, normalized within the current batch or pool, and transformed by a smooth gating function to produce the final weights $\widehat{w}_t(i)$. To ensure stability, weights are clipped to prevent excessively large effective learning rates and avoid gradient explosion. We now describe our score functions.

### 5.2 Model-Agnostic Per-Sample Learning Rate Update

**ADAPT-BM25** quantifies term-based overlap between training and validation data through sparse retrieval scoring:

$$s_{\text{BM25}}(x) = \frac{1}{|\mathcal{D}_{\text{val}}|} \sum_{v \in \mathcal{D}_{\text{val}}} \text{BM25}(x, v). \tag{8}$$

This metric captures surface-level textual similarity without semantic understanding. We use the standard BM25 score $s_{\text{BM25}}(v, i)$ over sparse token matches between query $v$ and example $i$. Aggregated scores are normalized to obtain target weights.

### 5.3 Per-Sample Learning Rate Update with Model States

**ADAPT** measures alignment using the model's own dense representations rather than a frozen encoder (Ivison et al., 2025). For an input $x$ with last-layer hidden states $\{h_i\}_{i=1}^L$, we compute a position-weighted mean pooling:

$$w_i = \frac{i}{\sum_{j=1}^L j}, \qquad \phi(x) = \sum_{i=1}^L w_i h_i, \tag{9}$$

where later tokens receive higher weights to counteract the causal mask bias of decoder-only models. We then define the similarity score:

$$s_{\text{ADAPT}}(x) = \frac{1}{|\mathcal{D}_{\text{val}}|} \sum_{v \in \mathcal{D}_{\text{val}}} \cos\big(\phi(x), \phi(v)\big). \tag{10}$$

Equivalently, we can instantiate a representation-based scorer $s_{\text{ADAPT}}(v, i) = \cos\big(\phi(v), \phi(i)\big)$, where $\phi(\cdot)$ is the weighted hidden representation.

We use a sigmoid function with temperature scaling to produce *absolute* weights that are independent of the batch composition:

$$w_t(i) = \sigma\left(\frac{s_{ADAPT}(x_i)}{\max(\tau, \epsilon)}\right) = \frac{1}{1 + \exp(-s_{ADAPT}(x_i)/\max(\tau, \epsilon))} \tag{11}$$

where $\sigma(\cdot)$ denotes the sigmoid function, $\tau > 0$ is a temperature hyperparameter (default $\tau = 1.0$), and $\epsilon > 0$ is a small constant (e.g., $\epsilon = 10^{-8}$) used for numerical stability. The temperature parameter $\tau$ controls the steepness of the sigmoid: larger values produce flatter weight distributions, while smaller values create sharper distinctions between high and low similarity samples.

**Global Weighting via Sigmoid:** The sigmoid transformation maps similarity scores to absolute weights in the interval $[0, 1]$ without requiring normalization across samples in the batch. This design choice ensures that the weight assigned to each sample $x_i$ depends solely on its similarity score $s_{ADAPT}(x_i)$ relative to the anchor set, rather than its rank within the current mini-batch. Consequently, a sample with a given similarity score receives the same weight regardless of whether it appears in a high-quality or low-quality batch, making the weighting mechanism robust to batch-level variations in data quality. This contrasts with normalized weighting schemes (e.g., softmax normalization) where weights are relative to other samples in the same batch, potentially amplifying or dampening effects based on batch composition.

**Embedding Normalization:** We apply L2 normalization to embeddings $\phi(x)$ and $\phi(v)$ before computing cosine similarities to ensure scale-invariant similarity measurements. Specifically, for each embedding vector, we compute $\phi(x) \leftarrow \phi(x)/\max(\|\phi(x)\|_2, \epsilon)$ where $\epsilon$ prevents division by zero. This normalization step is distinct from weight normalization and serves to standardize the embedding space for similarity computation.

**Online Validation Embedding Updates:** To maintain alignment with the evolving model representations, anchor embeddings $\{\phi(v)\}_{v \in \mathcal{D}_{val}}$ are refreshed every $R$ training steps via forward passes on the current model parameters $\theta_t$. This online update mechanism ensures that similarity scores reflect the model's current representation space rather than outdated embeddings from earlier training stages. The refresh interval $R$ is a hyperparameter that balances computational efficiency with representation freshness. The complete algorithm is summarized in Algorithm 1.

While gradient-based methods (e.g., LESS) directly measure influence, gradients significantly fluctuate during early training and are computationally expensive to compute frequently. In contrast, semantic embeddings provide a smoother, more stable signal of relevance that evolves consistently with the model's representation space, offering a robust guide for reweighting without the noise of instantaneous gradient estimates.

## 6 Evaluation

### 6.1 Experimental Design

**Instruction Tuning Models and Data.** We use LoRA (Hu et al., 2021) to fine-tune the base model LLAMA-2-7B (Touvron et al., 2023c). Following the experimental setup in Wang et al. (2023), we use the instruction tuning datasets including FLAN V2 (Longpre et al., 2023), CoT (Wei et al., 2022b), DOLLY (Conover et al., 2023) and OPEN ASSISTANT 1 (Köpf et al., 2023). The datasets do not contain any obvious in-domain data for the target queries. We evaluate our method on MMLU (Hendrycks et al., 2020), multiple-choice dataset spanning 57 tasks and BBH (Suzgun et al., 2023) from BIG-Bench selected to evaluate reasoning capabilities. For MMLU, we report 5-shot accuracy; for BBH, we report 3-shot exact match score. Appendix G contains more details on the training hyperparameter, Appendix D and E contain more dataset details.

**Instruction Tuning Baselines.** We introduce instruction finetuning baselines in Sec. 2.2.

**Pretraining Models and Data.** We adopt **Tinyllama** architecture (Zhang et al., 2024) with 120M parameters, with FlashAttention (Dao et al., 2022) and Lit-GPT (LightningAI, 2023). More details about the training settings can be found at Appendix F. Following prior works (Touvron et al., 2023b; Zhang et al., 2024; Wettig et al., 2024; Xie et al., 2023b), we employ **SlimPajama** (Touvron et al., 2023b; Computer, 2023) as the text corpus, which is specifically curated for pre-training LLMs. All selections are performed on about 590M training files of SlimPajama, processed with Llama tokenizer (Touvron et al., 2023b). We evaluate our method on a diverse set of 15 downstream benchmarks, following common practice in prior work such as RegMix (Liu et al., 2024). These tasks span a wide range of realistic settings, including: ARC-E (Clark et al., 2018), ARC-C (Clark et al., 2018), COPA (Sarlin et al., 2020), HellaSwag Zellers et al. (2019), Lambda-S (Paperno et al., 2016), Lambda-O (Radford et al., 2019), LogiQA (Liu et al., 2020), MultiRC (Khashabi et al., 2018), Open-

BookQA (Mihaylov et al., 2018), PiQA (Bisk et al., 2020), QQP (Wang, 2018), RACE (Lai et al., 2017), SciQ (Welbl et al., 2017), Social IQA (Sap et al., 2019), WinoGrande (Sakaguchi et al., 2021). The reported accuracy in table 2 is measured in the 0-shot setting scored using the lm-eval-harness evaluation framework (Gao et al., 2024).

**Pretraining Baselines.** We compare ADAPT with **Uniform** selection and existing file selection methods for LLM pre-training, including **Doremi** (Xie et al., 2023b) and **RegMix** (Liu et al., 2024). ADAPT requires an anchor set typically consists of examples in the evaluation distribution. To construct this set, we sample 50 validation examples from each of eight evaluation benchmarks: ARC-C, COPA, Lambada, MultiRC, PiQA, RACE, SciQ, and Social IQA. For Doremi and Regmix we use the domain weights in (Lu et al., 2023) as the selection ratio of text samples in different domains in our experiment. For Uniform we use the same ratio to sample from each domain. We also compare **LinUpper** (Sow et al., 2025): an online sample reweighting strategy where the sample weight is proportional to the normalized loss but is capped at a predefined $\alpha$ value, ensuring that outliers do not dominate the training process. Due to the large cost of pretraining, for each method we sample 9B unique tokens from SlimPajama, and train for a total budget of 50B tokens, i.e., train for approximately 5.6 epochs.

## 6.2 Results for Instruction Tuning

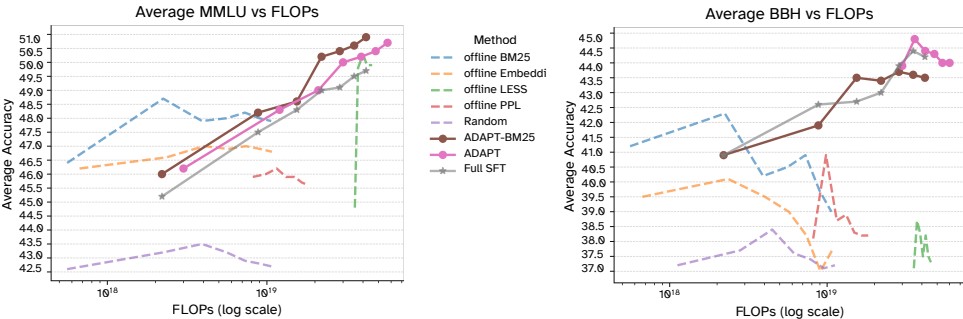

*(a)* In-domain evaluation: MMLU validation set → MMLU benchmark.

*(b)* Out-of-domain evaluation: MMLU validation set → BBH benchmark.

*Figure 2.* Efficiency-accuracy tradeoff. (a) While offline BM25 is efficient at low FLOPs, it saturates early. ADAPT dominates in the high-performance regime (higher FLOPs), avoiding the early plateauing of heuristics. (b) Crucially, LESS overfits to the in-domain task, showing regression on OOD tasks (BBH), whereas ADAPT consistently improves generalization on the Pareto frontier.

As shown in Figure 2, our proposed online data reweighting method consistently outperforms existing approaches under comparable computational budgets:

*Table 1.* Generalization results under different validation-test configurations.

| Method | MMLU*(val)* - MMLU*(test)* | MMLU*(val)* - BBH*(test)* |
|---|---|---|
| BM25 | 48.7 ± 0.9 | 42.3 ± 0.8 |
| Embedding | 47.0 ± 0.6 | 40.1 ± 0.5 |
| LESS | 50.2 ± 0.5 | 38.7 ± 1.5 |
| PPL | 46.2 ± 1.1 | 40.9 ± 0.9 |
| Random | 43.5 ± 0.3 | 38.4 ± 1.0 |
| Full Dataset SFT | 49.7 ± 0.2 | 44.4 ± 0.3 |
| ADAPT-BM25 | **50.9 ± 0.6** | **43.7 ± 1.2** |
| **ADAPT** | **50.7 ± 0.7** | **44.8 ± 1.3** |

**Comparison with offline data selection methods.** Across all baselines in both Figure 2a and Figure 2b, our approach consistently lies on the Pareto frontier, demonstrating the best trade-off between computational efficiency and final accuracy. At the same FLOPs, our method achieves substantially higher accuracy than the offline data selection SoTA method LESS. To reach the same performance level, our approach requires significantly fewer FLOPs than LESS. Moreover, after reaching this

*Table 2.* **Benchmark performance of *TinyLlama-120M* trained on 50B tokens using Uniform, LinUpper, DoReMi, RegMix, and ADAPT.**

| Tasks | Uniform | LinUpper | DoReMi | RegMix | ADAPT-BM25 | ADAPT |
|---|---|---|---|---|---|---|
| ARC-C | $17.75_{\pm0.00}$ | $18.77_{\pm0.24}$ | $18.00_{\pm0.84}$ | $18.60_{\pm0.06}$ | $18.31_{\pm0.55}$ | $19.11_{\pm0.60}$ |
| ARC-E | $39.60_{\pm0.10}$ | $38.85_{\pm0.12}$ | $40.70_{\pm0.87}$ | $41.04_{\pm0.21}$ | $39.44_{\pm0.60}$ | $39.06_{\pm0.54}$ |
| COPA | $63.00_{\pm0.00}$ | $60.00_{\pm1.72}$ | $66.00_{\pm1.95}$ | $61.00_{\pm1.5}$ | $63.20_{\pm0.40}$ | $64.00_{\pm1.32}$ |
| HellaSwag | $28.39_{\pm0.00}$ | $27.90_{\pm0.13}$ | $28.03_{\pm0.03}$ | $27.63_{\pm0.03}$ | $27.96_{\pm0.07}$ | $28.11_{\pm0.00}$ |
| Lambada-O | $24.68_{\pm0.64}$ | $23.64_{\pm0.28}$ | $22.38_{\pm0.73}$ | $24.82_{\pm0.94}$ | $26.19_{\pm0.30}$ | $24.63_{\pm0.53}$ |
| Lambada-S | $16.98_{\pm0.81}$ | $16.79_{\pm0.91}$ | $16.30_{\pm1.05}$ | $18.30_{\pm0.14}$ | $18.43_{\pm0.33}$ | $18.07_{\pm0.79}$ |
| LogiQA | $20.43_{\pm0.54}$ | $21.35_{\pm0.03}$ | $21.51_{\pm0.11}$ | $20.89_{\pm0.07}$ | $20.52_{\pm0.69}$ | $21.66_{\pm0.69}$ |
| MultiRC | $56.68_{\pm0.00}$ | $50.74_{\pm0.29}$ | $48.99_{\pm2.57}$ | $56.70_{\pm0.93}$ | $56.53_{\pm0.44}$ | $55.67_{\pm1.11}$ |
| OpenBookQA | $15.80_{\pm0.40}$ | $14.00_{\pm0.18}$ | $15.20_{\pm0.60}$ | $18.20_{\pm0.50}$ | $15.40_{\pm0.72}$ | $14.20_{\pm0.43}$ |
| PIQA | $60.55_{\pm0.29}$ | $60.83_{\pm0.45}$ | $59.90_{\pm0.39}$ | $58.98_{\pm0.29}$ | $61.00_{\pm0.19}$ | $61.48_{\pm0.23}$ |
| QQP | $36.88_{\pm0.00}$ | $37.85_{\pm1.21}$ | $36.84_{\pm0.04}$ | $36.83_{\pm0.00}$ | $36.82_{\pm0.00}$ | $36.81_{\pm0.00}$ |
| RACE | $27.85_{\pm0.14}$ | $26.41_{\pm0.18}$ | $26.41_{\pm0.03}$ | $27.85_{\pm0.00}$ | $27.39_{\pm0.35}$ | $26.60_{\pm0.49}$ |
| SciQ | $71.00_{\pm1.00}$ | $70.70_{\pm1.34}$ | $72.90_{\pm0.23}$ | $71.20_{\pm0.50}$ | $72.48_{\pm0.44}$ | $72.50_{\pm1.37}$ |
| SocialIQA | $36.90_{\pm0.03}$ | $36.85_{\pm0.16}$ | $36.95_{\pm0.07}$ | $37.41_{\pm0.46}$ | $36.52_{\pm0.47}$ | $37.05_{\pm0.30}$ |
| WinoGrande | $50.75_{\pm0.17}$ | $50.83_{\pm0.09}$ | $49.64_{\pm0.79}$ | $50.12_{\pm0.72}$ | $50.61_{\pm1.25}$ | $50.99_{\pm0.00}$ |
| Average (All) (↑) | $37.81_{\pm0.13}$ | $37.03_{\pm0.12}$ | $37.32_{\pm0.15}$ | $37.97_{\pm0.02}$ | $38.05_{\pm0.21}$ | $38.00_{\pm0.22}$ |
| Average (Unseen) (↑) | $31.98_{\pm0.09}$ | $30.56_{\pm0.28}$ | $31.95_{\pm0.17}$ | $32.46_{\pm0.39}$ | $33.49_{\pm0.37}$ | $33.73_{\pm0.39}$ |
| FLOPs overhead (↓) | 0 | 0 | $4.92 \times 10^{19}$ | $3.072 \times 10^{18}$ | $\ll 1.0 \times 10^{14}$ | $\ll 1.1 \times 10^{15}$ |

performance, further training with LESS leads to overfitting and accuracy degradation, whereas online data reweighting continues to exhibit steady performance gains. Notably, our online reweighting method using BM25 as quality signal (ADAPT-BM25) has better generalization than its offline selection method (BM25) with the other configuration keep the same, which demonstrates the advantage of the online data curation.

**Comparison with full-data fine-tuning.** In Figure 2a and Table 1, our method also outperforms full-data fine-tuning under equal FLOPs, with the performance gap widening as training progresses. In Fig. 2b and Table. 1, our reweighting strategy that incorporates model state information (ADAPT) outperforms our fixed-state online reweighting (ADAPT-BM25), highlighting the importance of adapting to the evolving model state to better generalisation.

**Generalization cross benchmark.** As shown in Figure 2b, we further validate the generalization capability of models trained with ADAPT, where we use MMLU (Hendrycks et al., 2021) as the validation set to select the data, and evaluate the model trained on selected data on the BBH (Suzgun et al., 2022) task. The results, also reported in Table 1, show that ADAPT achieves a better generalization performance between benchmark compared to other offline data selection baselines and demonstrates comparable generalization with vanilla full dataset instruction tuning. This highlights a key strength of ADAPT: when new downstream tasks or benchmarks emerge, there is no need to repeat the entire data selection and model training pipeline. Over the long term, this substantially reduces FLOPs costs, making ADAPT highly practical for real-world applications.

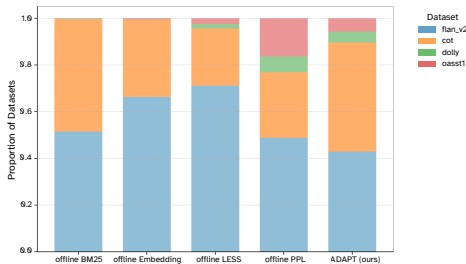

*(a)* Distribution of effective selected examples

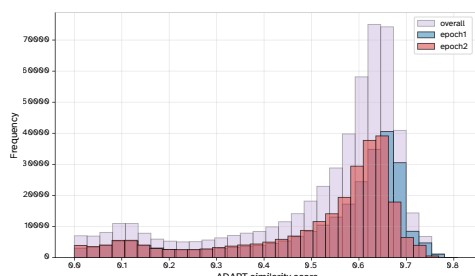

*(b)* Similarity distribution for different epochs

*Figure 3.* A summary of the distribution of effective selected samples and similarity scores. Both differences and changes show the proposed method's ability on capturing data features and adaptive learning.

**Data mixing and curriculum.** We present the distribution of the effective selected examples for different methods in Figure 3a. Different approaches end up selecting very different amounts of data from each training subset. It demonstrats that online reweighting solution can also help with deciding proper data mixture. We also calculate the *effective proportion of used data*, by summing up weights of all data together, with the result of 0.501 in our experimental setting. In practice, this number would automatically adapt according to the overall quality of the training corpus we are selecting from.

We also presented Figure 3b. The shift in similarity distributions from epoch 1 to epoch 2 reflects a transition from collapsed, overly homogeneous representations toward more diverse and fine-grained embeddings. This diversification enhances generalisation by improving the model's ability to capture subtle distinctions and reducing over-reliance on coarse features. Notably, this process can be viewed as an instance of implicit curriculum learning: the model first clusters samples based on coarse, "easier" patterns and progressively moves toward harder, fine-grained discrimination, thereby mirroring the principles of curriculum learning without explicit scheduling.

## 6.3 PRETRAINING RESULTS

As shown in Table 2, ADAPT outperforms all baselines in terms of average performance, achieving a 0.19% improvement over Uniform, 0.68% over DoReMi, and a 0.97% over LinUpper. RegMix reaches comparable but slightly lower accuracy on average. Interestingly, LinUpper performs worse than Uniform sampling, suggesting that naive sample-level adjustments may not generalize well. Compared to LinUpper, which conducts normalization and calculates sample-level weights for each batch, our global sample-level reweighting solution demonstrates superior performance in handling unknown ratios of mixed-quality corpora.

To examine the effect of larger training budgets, we extend training to 100B tokens. As shown in Table 3, ADAPT achieves higher benchmark performance than Uniform on 11 out of 15 downstream tasks, with an average improvement of 0.38%. Table 6 further shows that ADAPT consistently yields lower validation perplexity than Uniform at both 50B and 100B training budgets. These findings demonstrate that ADAPT improves both validation perplexity and downstream task performance consistently across compute scales.

*Table 3.* **Benchmark performance of *TinyLlama-120M* trained on 100B tokens using Uniform and ADAPT.**

| Tasks | Uniform | ADAPT |
|---|---|---|
| ARC-C | 17.75 | **18.86** |
| ARC-E | **40.66** | 40.45 |
| COPA | **63.00** | 61.00 |
| HellaSwag | 28.35 | **28.75** |
| Lambda-O | 25.13 | **25.91** |
| Lambda-S | 18.47 | **20.14** |
| LogiQA | 20.28 | **21.66** |
| MultiRC | 55.98 | **56.44** |
| OpenBookQA | **16.00** | 15.00 |
| PIQA | 61.48 | **61.97** |
| QQP | 36.81 | **36.82** |
| RACE | 27.37 | **27.46** |
| SciQ | 71.50 | **72.40** |
| SocialIQA | **35.77** | 37.05 |
| WinoGrande | 51.14 | **51.54** |
| Average (All) (↑) | 37.98 | **38.36** |
| Average (Unseen) (↑) | 32.51 | **32.67** |

## 7 CONCLUSION AND DISCUSSION

Our work unifies data selection, mixing, and reweighting under a FLOPs-aware framework. We show that offline pipelines often incur significant cost via data preprocessing and faces issues with generalization. In contrast, online reweighting integrates seamlessly into training, adapts to model state, and improves efficiency without affecting effective data size.

Empirically, we demonstrate that ADAPT consistently outperforms both offline and online baselines across instruction tuning and pretraining setups. In instruction tuning, ADAPT not only achieves higher accuracy under equal FLOPs but also exhibits stronger cross-benchmark generalization (e.g., from MMLU to BBH), mitigating the brittleness of offline methods. In pretraining, ADAPT improves both downstream task accuracy and validation perplexity under 50B and 100B token budgets, underscoring its robustness across compute scales. We further discuss the implications and outline future directions in Sec. A.

## Ethics and Reproducibility Statements

We adhere to the ICLR Code of Ethics. This paper focuses on comparing methodologies for data curation. To ensure fairness, we evaluate online data reweighting and offline data mixing methods under a FLOPs-equivalent setting. We also rigorously test their performance in both instruction fine-tuning and pretraining scenarios to provide a comprehensive and balanced comparison. During the preparation of this manuscript, we utilized large language models (LLMs) to assist with grammar correction and refinement of the writing. We provide all necessary details to ensure reproducibility of our work. The theoretical justification for our FLOPs-equivalent data curation formalization is presented in Section 3, with detailed FLOP calculations in Appendix C. Implementation details and training protocols are provided in Section 6.1, Appendix B, Appendix F, and Appendix G, while descriptions of the training data are given in Appendix D.

## Acknowledgement

We would like to thank Colin Raffel, Zhihao Zhan, Ziqing Fan, Jonathan Hayase, Guilherme Penedo, Hynek Kydlíček and Loubna Ben allal for many stimulating and helpful discussions. This research was supported by the following entities: The Royal Academy of Engineering via DANTE (a RAEng Chair); the European Research Council, specifically the REDIAL project; SPRIND under the composite learning challenge; Google through a Google Academic Research Award. Yihong acknowledges funding from the Horizon Europe grant No. 101213369 (DVPS).

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

# Contents

## A  IMPLICATION AND FUTURE WORK.

**Implication.** ADAPT could be effectively used for getting better control under constrained budgets (both *data budget* and *compute budget*). Our empirical observation provides a principled view and reference for automatically allocating computational resources to the most valuable samples while maintaining end-to-end efficiency. Additionally, we expect our method to perform even better where the data quality is often low, which is the common case in practical setup. where data is crude and is not ideal for direct use for LLM training, sources of generally low quality, and biases inherent to the distribution of content on the web.

**Future Work.** We didn't explicitly discuss and evaluate deduplication, which is a key step for data preprocessing. For example, we will treat deduplication as reweighting related documents in accordance with their frequency. Based on this framework, we will explore different trade-offs inherent in each stage, such as quality-aware deduplication where high-quality documents are allowed to be duplicated more than once.

In future work, we aim to explore ADAPT for safety-oriented data curation (Gal & Casper, 2025) and to study how dynamic data reweighting influences memorization and internal knowledge structure formation in large language models (Chen, 2025). In particular, combining ADAPT with interpretability techniques such as influence functions (Basu et al., 2020) and Jet Expansions (Chen et al., 2026) may enable us to trace how safety-critical data shapes learned representations and to examine whether models preserve underlying data distributions under dynamic reweighting. More broadly, we view ADAPT as a mechanism for dynamically modulating what information is reinforced during training. Such *data-centric* control of model memory opens avenues not only for improving model quality and safety but also for gaining insights into LLM internal knowledge formation when paired with model-side interpretability methods.

## B  INSTRUCTION TUNING BASELINES

We compare ADAPT with the following baselines: **1) Random Selection**: We randomly sample data from the instruction tuning dataset. **2) BM25** (Robertson et al., 2009): We assign weights to training samples based on textual statistical features (i.e., TF-IDF), and select the top $k$ data points with the highest scores. **4) PPL** (Yin & Rush, 2024; Marion et al., 2023; Ankner et al., 2024): We compute the loss of each training sample on our original base model and use it as its score. **6) LESS** Xia et al. (2024): We train LoRAs on a random subset of the data, and then selecting data by computing the gradient-based influence of each training sample to validation samples.

## C  DATA-SELECTION FLOPS

To estimate computational costs throughout our paper, we adopt the methodology of Kaplan et al. (2020b), which approximates the training step computation as approximately $6N$ FLOPs per processed token, where $N$ represents the model's parameter count (approximately 7B). According to Kaplan et al. (2020b), the forward pass consumes roughly half the computational resources of the backward pass, yielding an estimate of $2N$ FLOPs per token during sample processing. We employ an approximation of 2,048 tokens per sample, as we limit all samples to this maximum length during both the training and selection phases. Note that in all experimental configurations, we conduct full fine-tuning of models over two complete epochs. If we define $N$ as model size, $P$ as the data pool magnitude (measured in sample quantity), and $D$ as the number of samples chosen for training, we can calculate the computational expense for each methodology as follows:

1. **Random Selection**: $2k \times 6N \times D \times E$

2. **BM25**: $2k \times 6N \times D \times E$

3. **Embedding**: $2k \times 2N \times P + 2k * 6N \times D \times E$ (embed model $N$)

4. **PPL**: $2k \times 2NP + 2k \times 6N \times D \times E$

5. **LESS**: $1.53 \times 2k \times 6N \times P + 2k \times 6N \times D \times E$ (LESS computes gradients for three checkpoints over the entire pool.)

## D  Training Datasets in Instruction Tuning

Table 4 contains information about the training sets used in instruction tuning.

*Table 4.* Details of training dataset from Wang et al. (2023). Len. is short for token length.

| Dataset | # Instance | Sourced from | # Rounds | Prompt Len. | Completion Len. |
|---|---|---|---|---|---|
| FLAN V2 | 100,000 | NLP datasets and human-written instructions | 1 | 355.7 | 31.2 |
| CoT | 100,000 | NLP datasets and human-written CoTs | 1 | 266 | 53.2 |
| DOLLY | 15,011 | Human-written from scratch | 1 | 118.1 | 91.3 |
| OPEN ASSISTANT 1 | 55,668 | Human-written from scratch | 1.6 | 34.8 | 212.5 |

## E  Evaluation Datasets in Instruction Tuning

Table 5 contains detailed statistics of the evaluation datasets used in instruction finetuning.

*Table 5.* Statistics of evaluation datasets. The selection of evaluation tasks cover different kinds of answer types.

| Dataset | # Shot | # Tasks | $|\mathcal{D}_{val}|$ | $|\mathcal{D}_{test}|$ | Answer Type |
|---|---|---|---|---|---|
| MMLU | 5 | 57 | 285 | 18,721 | Letter options |
| BBH | 3 | 23 | 69 | 920 | COT and answer |

## F  Pretraining Details

We follow all settings in TinyLlama (Zhang et al., 2024). The optimizer is AdamW (Loshchilov & Hutter, 2019), setting parameters $\beta_1$ at 0.9 and $\beta_2$ at 0.95. We adopt the cosine learning rate schedule with a maximum learning rate of 4e-4 and the minimum of 4e-5, the batch size of 2M tokens, the weight decay of 0.1, and the gradient clipping threshold of 1.

## G  Instruction Tuning Details

All experiments were conducted with parameter-efficient finetuning method LoRA (Hu et al., 2021). For the LoRA adapter, we specified a rank of 128, an $\alpha$ value of 512, and a dropout rate of 0.1 and applied it across all attention matrices. Adding the LoRA adapter introduce minimal FLOPs overhead during training—having no impact on our FLOPS analysis—and mainly reduce memory requirements for more accessible training.

## H  Pretraining Perplexity

In Table 6, we evaluate validation perplexity on SlimPajama for TinyLlama-120M under two compute budgets (50B and 100B tokens). Across both settings, ADAPT achieves consistently lower perplexity than the Uniform baseline, highlighting its effectiveness in improving data efficiency.

*Table 6.* **Validation perplexity of TinyLlama-120M on SlimPajama under different training budgets.**

| Method | 50B Train Budget | 100B Train Budget |
|---|---|---|
| Uniform | 17.15 | 15.94 |
| ADAPT | 16.55 | 15.36 |

# I  PSEUDO CODE

---

**Algorithm 1:** Online Data Reweighting via Embedding Similarity

---

**Input** : Training set $\mathcal{D} = \{(x_i, y_i)\}_{i=1}^{N}$, validation set $\mathcal{D}_{val} = \{v_j\}_{j=1}^{|\mathcal{D}_{val}|}$, model $f_\theta$, temperature $\tau$, learning rate $\eta$, refresh interval $R$, training steps $T$, small constant $\epsilon > 0$

**Output** : Updated parameters $\theta_T$

1  Initialize model parameters $\theta_0$

2  **for** $t = 1$ **to** $T$ **do**

3 $\quad$ Sample mini-batch $B_t \subset \mathcal{D}$

$\quad$ ▷Refresh anchor embeddings every $R$ steps via forward pass on current model $\theta_t$ (online update)

4 $\quad$ **if** $t \bmod R = 1$ **or** $t = 1$ **then**

5 $\quad\quad$ **foreach** $v \in \mathcal{D}_{val}$ **do**

6 $\quad\quad\quad$ Forward pass: $(h_1^{(v)}, \ldots, h_{L_v}^{(v)}) \leftarrow f_{\theta_t}(v)$

7 $\quad\quad\quad$ Position weights: $w_\ell \leftarrow \ell / \sum_{j=1}^{L_v} j$ for $\ell = 1, \ldots, L_v$

8 $\quad\quad\quad$ $\phi(v) \leftarrow \sum_{\ell=1}^{L_v} w_\ell h_\ell^{(v)}$

9 $\quad\quad\quad$ $\phi(v) \leftarrow \phi(v)/\max(\|\phi(v)\|_2, \epsilon)$

10 $\quad\quad$ **end**

11 $\quad$ **end**

$\quad$ ▷Forward propagation for training samples

12 $\quad$ **foreach** $(x_i, y_i) \in B_t$ **do**

13 $\quad\quad$ Forward pass: $(\hat{y}_i, h_1^{(i)}, \ldots, h_{L_i}^{(i)}) \leftarrow f_{\theta_t}(x_i)$

14 $\quad\quad$ Position weights: $w_\ell^{(i)} \leftarrow \ell / \sum_{j=1}^{L_i} j$ for $\ell = 1, \ldots, L_i$

15 $\quad\quad$ $\phi(x_i) \leftarrow \sum_{\ell=1}^{L_i} w_\ell^{(i)} h_\ell^{(i)}$

16 $\quad\quad$ $\phi(x_i) \leftarrow \phi(x_i)/\max(\|\phi(x_i)\|_2, \epsilon)$

17 $\quad$ **end**

$\quad$ ▷Compute quality scores

18 $\quad$ **foreach** $i \in B_t$ **do**

19 $\quad\quad$ $s_{ADAPT}(x_i) \leftarrow \frac{1}{|\mathcal{D}_{val}|} \sum_{v \in \mathcal{D}_{val}} \cos(\phi(x_i), \phi(v))$

20 $\quad$ **end**

$\quad$ ▷Compute per-sample weights

21 $\quad$ **foreach** $i \in B_t$ **do**

22 $\quad\quad$ $w_t(i) \leftarrow \sigma(s_{ADAPT}(x_i)/\max(\tau, \epsilon))$

23 $\quad$ **end**

$\quad$ ▷Compute weighted loss

24 $\quad$ $\mathcal{L}_t \leftarrow \sum_{i \in B_t} w_t(i)\ell(\hat{y}_i, y_i)$

$\quad$ ▷Backward propagation and parameter update

25 $\quad$ $\theta_{t+1} \leftarrow \theta_t - \eta \sum_{i \in B_t} w_t(i)\nabla_{\theta_t}\ell(\hat{y}_i, y_i)$

26 **end**

27 **return** $\theta_T$

---

# J  QUANTITATIVE ANALYSIS

We provide a detailed qualitative comparison in Table 7, Table 8 and Table 9.

*Table 7.* Examples with highest and lowest scores retrieved by BM25 for the MMLU validation set.

---

**MMLU Validation Set Examples**

---

**User:** Which expression is equivalent to 5 x 9? A. (5 x 4) x (6 x 5) B. (5 x 5) + (5 x 4) C. (5 x 5) + (5 x 9) D. (5 x 9) x (6 x 9). Answer:
**Assistant:** B

**User:** Let A be the set of all ordered pairs of integers (m, n) such that $7m + 12n = 22$. What is the greatest negative number in the set $B = \{m + n : (m, n) \in A\}$? A. -5 B. -4 C. -3 D. -2 Answer:
**Assistant:** B

**User:** Ten students take a biology test and receive the following scores: 45, 55, 50, 70, 65, 80, 40, 90, 70, 85. What is the mean of the students' test scores? A. 55 B. 60 C. 62 D. 65 Answer:
**Assistant:** D

---

| **Retrieved by BM25 (Highest similarity)** | **Retrieved by BM25 (Lowest similarity)** |
|---|---|

**Sample Index:** cot 60736
**Score:** 0.9403
**User:** Question: A multiple choice test consists of 4 questions, and each question has 5 answer choices. In how many ways can the test be completed if every question is unanswered? Options: (A) 24 (B) 120 (C) 625 (D) 720 (E) 1024 Answer: 5 choices for each of the 4 questions, thus total of $5 \times 5 \times 5 \times 5 = 5^4 = 625$ ways.
**Assistant:** The answer is (C).

**Sample Index:** cot 60736 (same entry, continued)
**Score:** 0.9403
**User:** Question: In how many different ways can the letters of the word "CLAIM" be rearranged? Options: (A) 110 (B) 120 (C) 125 (D) 130 (E) 140 Answer: The total number of arrangements is 5! = 120.
**Assistant:** The answer is (B).

**Sample Index:** cot 60736 (same entry, continued)
**Score:** 0.9403
**User:** Question: A pipe can fill a cistern in 20 minutes whereas the cistern when full can be emptied by a leak in 70 minutes. When both are opened, find when the cistern will be full? Options: (A) 17 minutes (B) 87 minutes (C) 70 minutes (D) 28 minutes (E) 11 minutes Answer: $1/20 - 1/70 = (7 - 2)/140 = 5/140 = 1/28$   28 minutes.
**Assistant:** The answer is (D).

**Sample Index:** cot 60736 (same entry, continued)
**Score:** 0.9403
**User:** Question: What is the area of a rectangle whose length is twice its width and whose perimeter equals that of a square with area 1? Options: (A) 1/9 sq units. (B) 4/9 sq units. (C) 8/9 sq units. (D) 7/9 sq units. (E) 6/9 sq units. Answer: Perimeter of square = 4, so perimeter of rectangle = $4 \to 2(l + w) = 4 \to l + w = 2$. Since $l = 2w \to 3w = 2 \to w = 2/3$, $l = 4/3$. Area = $(4/3) \times (2/3) = 8/9$.
**Assistant:** The answer is (C).

**Sample Index:** oasst1_41616
**Score:** 0.0
**User:** ทำไมภาษาลาวคล้ายกับภาษาไทย
**Assistant:** ภาษาลาวและภาษาไทยมีความคล้ายคลึงกันเนื่องจากเป็นภาษาในกลุ่มภาษาไทย-ลาว ซึ่งเป็นกลุ่มภาษาในภูมิภาคอินโดจีน ซึ่งประกอบด้วยภาษาที่มีลักษณะทางการเขียนเหมือนกันและมีความคล้ายคลึงกันในระบบไวยากรณ์และการออกเสียง ารที่ภาษาลาวและภาษาไทยมีความคล้ายคลึงกันเป็นเรื่องธรรมชาตินื่องจากมีประวัติศาสตร์และวัฒนธรรมที่คล้ายคลึงกัน โดยทั้งสองภาษามีผลมาจากการได้รับอิทธิพลจากภูมิภาคและประวัติศาสตร์เดียวกัน                   นอกจากนี้ การมีการติดต่อกันและการแลกเปลี่ยนทางวัฒนธรรมในประวัติศาสตร์ระหว่างประเทศลาวและไทยก็ส่งผลให้ภาษาลาวและภาษาไทยมีความคล้ายคลึงกันในบางด้านของภาษาด้วย

**Sample Index:** oasst1_4573
**Score:** 0.0
**Question:** Buenos días
**Answer:** Hola buenos días. ¿En qué te puedo ayudar?

**Sample Index:** oasst1_42983
**Score:** 0.0
**Question:** 第一次去深圳旅游，给我安排一下行程，大概3天的时间。
**Answer:** 深圳是中国广东省的一座现代化大城市，以其高新科技、美丽的海滨风光和丰富的购物娱乐场所而闻名。这里为您安排一个3天的深圳旅游行程：[...] (完整3天行程)

**Sample Index:** oasst1_43212
**Score:** 0.0
**Question:** 私は猫である名前はまだない
**Answer:** 夏目漱石の「吾輩は猫である」という作品の最初の一節だね。

---

*Table 8.* Examples with highest and lowest scores retrieved by LESS (gradient-based) for the MMLU validation set.

---

**MMLU Validation Set Examples**

---

**User:** Which expression is equivalent to 5 x 9? A. (5 x 4) x (6 x 5) B. (5 x 5) + (5 x 4) C. (5 x 5) + (5 x 9) D. (5 x 9) x (6 x 9). Answer:
**Assistant:** B

**User:** Let A be the set of all ordered pairs of integers (m, n) such that $7m + 12n = 22$. What is the greatest negative number in the set $B = \{m + n : (m, n) \in A\}$? A. -5 B. -4 C. -3 D. -2 Answer:
**Assistant:** B

**User:** Ten students take a biology test and receive the following scores: 45, 55, 50, 70, 65, 80, 40, 90, 70, 85. What is the mean of the students' test scores? A. 55 B. 60 C. 62 D. 65 Answer:
**Assistant:** D

---

| **Retrieved by LESS (Highest similarity)** | **Retrieved by LESS (Lowest similarity)** |
| --- | --- |

**Sample Index:** flan_v2_78815    **Score:** 0.9922

**User:** the compound ratio of 2/3 , 6/7 , 4/3 and 1/8 is given by? Options: a) 2/21 , b) 1/48 , c) 1/98 , d) 1/96 , e) 1/68
**Assistant:** a

**Sample Index:** flan_v2_81227    **Score:** 0.0

**User:** Personality: I only sleep 3 hours a day. I have no parents or siblings. I'm incredibly smart. [...] I don't blame you. What is your favorite food? I like well done steak. Candidates 1) Hi, how are you doing? 2) Hi, I am Kathie! I don't like eating fast food. 4) I like Mac N cheese.
**Assistant:** I like Mac N cheese.

**Sample Index:** flan_v2_40013    **Score:** 1.0

**User:** Look, this is a photo of Mr. White's family. The man in the photo is Mr. White, the father. The woman is the mother. They have two daughters. The girl behind Mr. White is Mary. She's 14. The girl behind Mrs. White is Jane. She is 11. Mary and Jane are in the same school, but not the same grade. Mary is in Grade2. Jane is in Grade 1. They are good students. How old are Mary and Jane?
Options: A) Eleven and thirteen B) Fourteen and eleven C) Eleven D) Thirteen
**Assistant:** B

**Sample Index:** flan_v2_40281    **Score:** 0.0187

**User:** Where will a book being read by one person be kept? Options: school room, university, bedside table, pantry, discount store I believe the correct choice is "bedside table", here's why:
**Assistant:** a book being read by one person be kept bedside table.

**Sample Index:** flan_v2_40832    **Score:** 0.946

**User:** Please answer the following question: Dogs have an understanding of fair play and become angry if they feel that another dog is getting a better deal, a new study has found. The study looked at how dogs react when a companion is rewarded for the same trick in an unequal way. Friederike Range, a researcher at the University of Vienna in Austria, and her colleagues did a series of experiments with dogs who knew how to respond to the command "give the paw ". The dogs were normally happy to repeatedly give the paw, whether they got a reward or not. Range found that, compared with monkeys, dogs _.
Options: A care more about whether they are rewarded. B care less about what they are rewarded with. C care more about what they are ordered to do. D care less about who gives them orders.
**Assistant:** B

**Sample Index:** flan_v2_78179    **Score:** 0.0153

**User:** sbjerg forenede Boldklubber (EfB) is a professional Danish football club from the port city of Esbjerg in West Jutland. The club was formed in 1924, as a merger between Esbjerg Boldklub af 1898 and Esbjerg Amatørklub af 1911. The club has training facilities and stadium at Gl. Vardevej in Esbjerg, and plays in blue and white striped shirts. [...] Subject: morten karlsen, Relation: place_of_birth. Options: *A* 1955 *B* central *C* copenhagen *D* denmark *E* esbjerg *F* farum *G* island (H) jutland (I) of (J) once (K) randers (L) time (M) uerdingen (N) union (O) viborg (P) zealand
**Assistant:** copenhagen

---

*Table 9.* Examples with highest and lowest scores retrieved by ADAPT for the MMLU validation set.

| MMLU Validation Set Examples |
|---|

**User:** Which expression is equivalent to 5 x 9? A. (5 x 4) x (6 x 5) B. (5 x 5) + (5 x 4)
C. (5 x 5) + (5 x 9) D. (5 x 9) x (6 x 9). Answer:
**Assistant:** B

**User:** Let A be the set of all ordered pairs of integers (m, n) such that $7m + 12n = 22$.
What is the greatest negative number in the set $B = \{m + n : (m, n) \in A\}$? A. -5 B. -4
C. -3 D. -2 Answer:
**Assistant:** B

**User:** Ten students take a biology test and receive the following scores: 45, 55, 50, 70,
65, 80, 40, 90, 70, 85. What is the mean of the students' test scores? A. 55 B. 60 C. 62
D. 65 Answer:
**Assistant:** D

| Retrieved by ADAPT (Highest similarity) | Retrieved by ADAPT (Lowest similarity) |
|---|---|
| **User:** Write something based on this summary: canada foreign minister regrets handling of visa denial for haitian prime minister [...]
**Assistant:** the maximum disturbing and unraveling word for an author 's agent to hear is "no." so in the ceremony of book publishing , the relationship that is most culturally challenging and polarizing is the one between writers ' agents and publishers. [...]

**User:** Generate a short summary: an airplane en route to algiers from paris made an unscheduled stop thursday in central france after a passenger reported mistakenly bringing on board a suitcase that belonged to someone else [...] (several summarization tasks)
**Assistant:** paris-algiers flight makes unexpected stop in central france [...] (several summarization answers) | **User:** Vale
**Assistant:** Dime Tio, que necesitas?

**User:** Write a sentence not in English.
**Assistant:** MDN DOM 195 Chef - Service d'examen Vérification Description:

**User:** Kiiltoraunioinen Translate to English
**Assistant:** Asplenium adiantum-nigrum |

