# OpenReview forum: "Rethinking Data Curation in LLM Training: Online Reweighting  Offers Better Generalization than Offline Methods"
_ICLR.cc/2026/Conference — ICLR 2026 Poster_

### Official Review · Reviewer_P23q · 2025-10-28

**Soundness:** 2
**Presentation:** 3
**Contribution:** 2
**Rating:** 2
**Confidence:** 3

**Summary:**

This paper propose an online reweight mechanics for LLM training samples. The author conduct various experiments to demonstrate the effectiveness of the algorithm.

**Strengths:**

1. The paper is easy to read and understand.
2. The idea is clear and easy to follow.

**Weaknesses:**

1. This paper claims a lot contribution of "online" as stated in the title, but there exist several online reweight algorithms. For example [1] and [2]. The fair comparison should be made between proposed algorithm and other online algorithms. The storytelling is misleading such comparison is absent.
2. The improvement is limited. As we can see in Table 2, ADAPT improves a little comparing against other baselines. For example, ADAPT (38.00) is a little better than RegMix (37.97). The improvement 0.03 might be under standard deviation range.
3. The experimental setting is limited. The author is encouraged to perform more experiments across different model sizes and architectures to test algorithm robustness.


[1] Importance weighting can help large language models self-improve
[2] Take the Bull by the Horns: Hard Sample-Reweighted Continual Training Improves LLM Generalization

**Questions:**

Please check the weakness part.

---

> ### Author Response · Authors · 2025-11-28
>
> We thank the reviewer for finding our paper easy to read and the idea clear. We value your feedback. We noticed that the summary and contribution assessment (Score: 2) primarily focused on the specific algorithm, potentially overlooking our broader contributions. We respectfully address your concerns regarding baselines, performance magnitudes, and robustness below.
>
> ---
>
> ## W1
>
> > This paper claims a lot contribution of "online" as stated in the title, but there exist several online reweight algorithms. For example [1] and [2]. The fair comparison should be made between proposed algorithm and other online algorithms. The storytelling is misleading such comparison is absent.
>
> We respectfully clarify that comparisons are **not absent**. We explicitly compared against a representative online sample reweighting method, LinUpper [Sow et al., 2025], in our experiments. As shown in Table 2, ADAPT (38.00) significantly outperforms LinUpper (37.32) on average accuracy.
>
> We acknowledge that [1], [2], and our paper all share the broad goal of data-centric optimization to enhance model performance. We have cited them in the revision. However, they operate under fundamentally different assumptions and settings compared to ours:
>
> * [1] focuses on Self-Improvement using synthetic, self-generated data. In contrast, ADAPT tackles the challenge of curating massive raw external corpora (e.g., SlimPajama). We deal with the intrinsic noise and distribution shifts of the open web, a fundamentally different data distribution problem than self-generated refinement.
> * [2] focuses on continual training by emphasizing samples with "moderately high loss." This methodology relies on loss signals from a pre-trained model and cannot be applied to training from scratch, where the model is initially random and loss signals are uninformative. In contrast, ADAPT is capable of pretraining from scratch (as shown in our TinyLlama experiments) as well as fine-tuning, offering a more versatile solution for the full LLM lifecycle.
> ---
>
> ## W2
>
> > The improvement is limited. As we can see in Table 2, ADAPT improves a little comparing against other baselines. For example, ADAPT (38.00) is a little better than RegMix (37.97). The improvement 0.03 might be under standard deviation range.
>
> We clarify this with updated Table 2 reporting standard deviations and FLOPs overhead.
>
> | Tasks                    | Uniform    | LinUpper   | DoReMi     | RegMix     |  ADAPT          |
> | ------------------------ | ---------- | ---------- | ---------- | ---------- | -------------- |
> | **Average (All) (↑)**    | 37.81±0.13 | 37.03±0.12 | 37.32±0.15 | 37.97±0.02 | **38.00±0.22**     |
> | **Average (Unseen) (↑)** | 31.98±0.09 | 30.56±0.28 | 31.95±0.15 | 32.46±0.39 | **33.73±0.39** |
> | **FLOPs overhead (↓)**   | 0                | 0                | 4.92×10¹⁹        | 3.072×10¹⁸       | ≪1.1×10¹⁵
>
> **Statistical Significance on Generalization**: We updated Tables 1 & 2 to report standard deviations across 3 independent runs. Crucially, ADAPT shows a decisive advantage on unseen tasks (tasks not used in the anchor/validation set), which shows the generalization capability. In the pretraining setting, ADAPT outperforms RegMix by 1.75%. This gap is **>3x the standard deviation**, confirming the gain is statistically significant and robust. Also, in fine-tuning setting, ADAPT outperforms the offline SOTA LESS on the out-of-domain BBH benchmark by 6.1%.
>
> **Decisive Advantage in Performance-FLOPs Trade-off**: ADAPT achieves these gains with orders of magnitude lower computational overhead. ADAPT requires less than 0.1% of the overhead required by RegMix, yet yields superior generalization. This supports our core claim that online reweighting offers a fundamentally better pareto frontier than offline methods.

---

> > ### Author Response · Authors · 2025-11-28
> >
> > ## W3
> >
> > > The experimental setting is limited. The author is encouraged to perform more experiments across different model sizes and architectures to test algorithm robustness.
> >
> > We respectfully highlight that our evaluation is rigorously conducted to test algorithm robustness across three distinct dimensions:
> >
> > 1. Tables 1–3 are based on 3 independent runs per method. In pretraining, we use seeds 0, 1, and 2. In fine-tuning, we use seeds 3, 6, and 9. For each run, we retrain the model and re-run evaluation. The reported standard deviations confirm ADAPT’s consistent superiority.
> >
> > 2. Unlike baselines limited to one stage, ADAPT is a unified framework and we conducted experiments for validation on both **pretraining** (TinyLlama-120M on SlimPajama) and **fine-tuning** (Llama-2-7B on four datasets).
> >
> > 3. The 120M scale is a standard academic proxy allowing for rigorous multi-seed verification. We have extended pretraining to 100B tokens, where ADAPT outperformed Uniform on average performance of 15 diverse tasks. To further validate scalability, we have launched pretraining experiments on a 1.1B model. Preliminary results align with our findings, and we are committed to including these in the camera-ready version.
> >
> > In addition, we have also conducted additional experiments and incorporated the detailed results into the revised manuscript. The table below summarizes the key ablation results in comparison to the baselines and our proposed ADAPT method:
> >
> > | Method     | Average (↑)     |
> > |------------|-----------------|
> > | Uniform    | 37.81±0.13      |
> > | LinUpper   | 37.03±0.12      |
> > | DoReMi     | 37.32±0.15      |
> > | RegMix     | 37.97±0.02      |
> > | ADAPT      | 38.00±0.22      |
> > | **ADAPT-MP** (mean pooling)   | **37.80±0.41**      |
> > | **ADAPT-SV** (smaller validation set)   | **37.98±0.34**      |
> > | **ADAPT-SP** (SlimPajama validation anchors)  | **37.97±0.27**      |
> > | **ADAPT-BM25**   | **38.05±0.21**      |
> >
> > (i) Replacing our weighted mean-pooling with a uniform mean-pooling (denoted as ADAPT-MP) led to a performance decrease from $38.00 \pm 0.22$ to $37.80 \pm 0.41$. This demonstrates the necessity and advantage of our weighted mean-pooling approach. The learned weights effectively capture the importance of representations at different training steps, leading to superior overall performance compared to unweighted averaging.
> >
> > (ii) We performed ablations on both the size and composition of the anchor set to demonstrate the robustness of our method.
> >
> > *Size:* We significantly reduced the anchor set size, sampling only 20 validation examples from each of the eight evaluation benchmarks, totaling 160 samples. The performance (shown as ADAPT-SV, smaller validation) was $37.98 \pm $. The performance fluctuated only slightly (a decrease of 0.02) and still significantly outperformed other baselines. This confirms the high stability of our proposed method with respect to the anchor-set size.
> >
> > *Composition:* We changed our anchor set to the validation set from SlimPajama (a general, unrelated corpus validation set). The performance (shown as ADAPT-SP, SlimPajama) was $37.97 \pm 0.27$. The performance again fluctuated only slightly (a decrease of 0.03), maintaining a strong result. This demonstrates the generalization and robustness of ADAPT, as it performs well even when the anchor set is sourced from a general-purpose corpus validation set rather than task-specific validation data.
> >
> > (iii) We replaced the model-state embedding with the BM25 score as the quality signal within the ADAPT framework. The result (shown as ADAPT-BM25) was $38.05 \pm 0.21$. Surprisingly, using BM25 as the quality signal not only outperformed all other baselines but also slightly surpassed our original model-state embedding approach ($38.05$ vs $38.00$). This highlights the flexibility and efficacy of the ADAPT framework, showing that it can successfully leverage diverse quality signals (both dense embeddings and sparse lexical metrics) to effectively modulate training. We have updated Table 2 to include detailed per-task results and FLOPs comparison.
> >
> > -----
> > In summary, ADAPT provides a principled, unified framework that addresses the critical generalization and efficiency limitations of offline methods. Given the statistically significant gains on unseen tasks, the unified application across training paradigms, and the decisive 1000× FLOPs efficiency advantage, we believe ADAPT makes a substantial contribution to efficient LLM training and respectfully ask the reviewer to reconsider the score.
> >
> > We sincerely hope that the above responses sufficiently address all your concerns. Should there be any inaccuracy in our understanding, or if you have any further questions on any detail, we would be very happy to provide additional clarification and discussion!

---

> ### Author Response · Authors · 2025-12-04
>
> We are updating to share more experimental results on different sizes of base models: Llama2-13B. The results are summarized in the table below:
>
> | Method         | Performance |
> | -------------- | ---------------------- |
> | Random         | 52.5 ± 0.6             |
> | **ADAPT-BM25** | **55.2 ± 0.6** |
> | **ADAPT** | **55.4 ± 0.5** |
>
> Consistent with our findings on Llama-2-7B, **ADAPT** significantly outperforms the Random baseline on the larger Llama-2-13B model by **2.9%**. This improvement is substantial and exceeds the standard deviation, confirming that our online reweighting framework is robust and scales effectively to larger model architectures. These results directly address your suggestion to "perform more experiments across different model sizes" and demonstrate that ADAPT's benefits are not limited to smaller scales.

---

### Official Review · Reviewer_osmP · 2025-10-29

**Soundness:** 3
**Presentation:** 3
**Contribution:** 3
**Rating:** 6
**Confidence:** 3

**Summary:**

This paper proposes ADAPT, a method for selecting influential data, selecting mixtures of data, and reweighting data -- all in one framework. They do this by defining a function $s(\cdot)$ that will assign a score to an input. In data selection, samples are selected by applying a threshold on the scoring function (only data that has a high score will be chosen). In data mixing across the domain level, the scoring function is used to determine how much weight a domain (cluster of samples, perhaps) should be given. They do this by transforming the score with the $exp(\cdot)$ function and then normalize the scores across domains. Finally, in (online) data reweighting, the scoring function is used to determine how much to scale the loss of the sample by during gradient descent. The paper shows competitive performance compared to baselines.

**Strengths:**

- The problem of unifying online and offline data curation is a timely problem
- The paper is well organized and written
- The experimentation is extensive and thorough

**Weaknesses:**

- According to Figure 2a, ADAPT-BM25 and ADAPT don't do much better than LESS, with the same number of FLOPs
- According the Figure 2a/2b, with fewer FLOPs, BM25 is able to beat most baselines and ADAPT. Which means that under a tighter FLOPs budget, a simpler method like BM25 could achieve best performance (and it also generalizes to different datasets well)
- I might have missed it, but I could not find any analysis on the qualitative results of data selection, mixing, and reweighting

**Questions:**

1. In the "Setup" paragraph in Section 3, lines 140-146: what do you mean by "domain"? If "in-domain" and "out-of-domain" are interchangeable with "in-distribution" and "out-of-distribution", why does $D_{train}$ contain in-domain _and_ out-of-domain. Since the model has access to the training set, doesn't that automatically mean that all training data is in-domain? If "domain" is not the same as "distribution", how do you define and determine "domain"?

---

> ### Author Response · Authors · 2025-11-28
>
> We thank the reviewer for recognizing our work as a "timely problem," our experimentation as "extensive and thorough", and the paper as "well organized." We address your specific questions below to further clarify the contributions.
>
> ------
>
> ## W1
> > According to Figure 2a, ADAPT-BM25 and ADAPT don't do much better than LESS, with the same number of FLOPs
>
> While LESS appears competitive on the In-Domain task (Fig. 2a), Figure 2b (OOD: MMLU $\to$ BBH) reveals a critical failure mode of offline selection: As training progresses, LESS suffers a significant performance drop on the unseen BBH benchmark.
>
> **Why this happens:** As detailed in our qualitative analysis (Response to W3), this is because LESS heavily overfits to the specific format of the validation set (Hard Filtering), discarding diverse data required for OOD tasks.
>
> **ADAPT's Advantage:** In contrast, ADAPT maintains a steady upward trajectory on both ID and OOD benchmarks, eventually outperforming LESS by a large margin (>5%) on BBH. This confirms that ADAPT's online soft-weighting preserves data diversity, preventing the brittleness inherent in LESS.
>
> > According the Figure 2a/2b, with fewer FLOPs, BM25 is able to beat most baselines and ADAPT. Which means that under a tighter FLOPs budget, a simpler method like BM25 could achieve best performance (and it also generalizes to different datasets well)
>
> The reviewer is correct that BM25 is efficient at very low FLOPs, but it suffers from **the ceiling effect: offline BM25** (blue dashed line) **saturates quickly**. ADAPT continues to improve with more FLOPs because it dynamically adjusts to the model's evolving state.
>
> ADAPT breaks such ceiling. While it incurs an initial compute cost, it continuously improves model performance beyond the limits of BM25. For high-performance model training, ADAPT represents a superior Pareto frontier, offering higher peak accuracy and robustness that simple heuristics cannot achieve.
>
> ## W3
> > I might have missed it, but I could not find any analysis on the qualitative results of data selection, mixing, and reweighting.
>
> We provided a detailed qualitative comparison in Appendix J (Tables 7, 8, and 9). We have now expanded this analysis to highlight critical findings:
>
> **BM25 (Lexical Bias)**: As shown in Table 7, BM25 retrieves samples with high surface-level keyword overlap. So it incorrectly filters out high-quality multilingual data (e.g., Thai and Chinese samples) simply because they lack keyword overlap with the English validation set. This limitation restricts the model's potential performance gain by discarding diverse linguistic signals.
>
> **LESS (Format Overfitting)**: Our analysis reveals that LESS tends to assign high scores to samples that strictly match the format of the validation set (Table 8). Consequently, it aggressively filters out high-quality data that has the correct knowledge but a different format. As seen in the "Lowest similarity" column of Table 8, LESS assigns near-zero scores to valuable open-ended reasoning samples. This "hard filtering" and "format bias" reduce data diversity, lead to the overfitting and explains its poor generalization to out-of-domain tasks like BBH, which includes non-multiple-choice reasoning (Figure 2b).
>
>
> **ADAPT (Semantic Quality & Diversity Balance)**: In contrast, ADAPT demonstrates a robust ability to identify semantically high-quality data regardless of format. As shown in Table 9, ADAPT assigns high weights to reasoning-intensive tasks like summarization (e.g., "Canada foreign minister..."), recognizing their utility for the model's general capabilities even though they do not match the multiple-choice format of the validation set. Also, ADAPT effectively identifies and down-weights **true noise**, such as meaningless short texts or fragmented sentences. By using **online soft-weighting** instead of hard filtering, ADAPT retains the entire training pool (preserving **diversity**) while dynamically emphasizing high-quality data (improving **quality**). This mechanism allows ADAPT to excel in-domain (MMLU) while maintaining the robustness required for out-of-domain generalization (BBH).

---

> ### Author Response · Authors · 2025-11-28
>
> ## Q:
> > In the "Setup" paragraph in Section 3, lines 140-146: what do you mean by "domain"? If "in-domain" and "out-of-domain" are interchangeable with "in-distribution" and "out-of-distribution", why does contain in-domain and out-of-domain. Since the model has access to the training set, doesn't that automatically mean that all training data is in-domain? If "domain" is not the same as "distribution", how do you define and determine "domain"?
>
> We clarify that our usage of "Domain" follows the standard conventions in data curation literature (e.g., LESS [Xia et al., 2024], DSIR [Xie et al., 2023b]), where the training set ($\mathcal{D}_{train}$) is treated not as a curated distribution, but as a large, heterogeneous raw data pool collected from mixed sources (e.g., web crawls).
>
> In this context, "In-Domain" and "Out-of-Domain" (OOD) are defined relative to the Validation Set ($\mathcal{D}_{val}$), which serves as our target quality anchor.
> * In-Domain (ID): Samples within the raw $D_{train}$ that are distributionally aligned with $\mathcal{D}_{val}$ (relevant signal).
> * Out-of-Domain (OOD): Samples within $D_{train}$ that are irrelevant or distributionally distinct from $\mathcal{D}_{val}$ (noise).
>
> This definition remains consistent throughout our experiments (e.g., Figures 1 & 2):
> * In-Domain Evaluation: The Test Set ($D_{test}$) shares the same source as the anchor $\mathcal{D}_{val}$ (e.g., MMLU val $\to$ MMLU test).
> * Out-of-Domain Evaluation: The Test Set ($D_{test}$) differs from the source of $\mathcal{D}_{val}$ (e.g., MMLU val $\to$ BBH test).
>
> This measures whether the method learns robust features (generalization) rather than simply memorizing the specific distribution of $\mathcal{D}_{val}$.
>
> ------
> We sincerely hope that the above responses sufficiently address all your concerns. If you have any further questions on any detail, we would be very happy to provide additional clarification and discussion!

---

### Official Review · Reviewer_SsJ4 · 2025-10-31

**Soundness:** 2
**Presentation:** 3
**Contribution:** 3
**Rating:** 4
**Confidence:** 3

**Summary:**

The paper argues that many existing “offline” data-curation pipelines (selection/mixing) are brittle and compute-inefficient when judged end‑to‑end, and proposes reframing curation as online sample reweighting integrated into training. The method, ADAPT, implements per‑sample learning‑rate multipliers via similarity‑based scores to a validation/anchor set, including a variant that uses the model’s own evolving representations (Eq. 7–10). Empirically, ADAPT is claimed to provide a better accuracy‑vs‑FLOPs trade-off than offline selection/mixing and prior online reweighting, on instruction tuning (Llama‑2‑7B with LoRA) and pretraining (TinyLlama‑120M on SlimPajama). Key visuals include the trade-off plots (MMLU/BBH vs FLOPs, Fig. 1 & Fig. 2) and tables showing small but consistent average gains (Tables 1–3)

**Strengths:**

- Unified view and cost accounting. The paper formalizes selection, mixing, and reweighting under one umbrella and compares them with a total FLOPs accounting that includes pre‑/in‑loop costs (Eqs. 5–6 on p.4; Sec. 3.4). This framing is valuable for the community, where “hidden” preprocessing costs are often ignored

- Simple, generally applicable mechanism. Casting curation as per‑sample learning‑rate scaling (Eq. 7) is elegant and easy to integrate in standard optimizers; using model-state representations (Eq. 9–10) to compute similarity is a sensible way to keep the weighting adaptive as training proceeds (Sec. 5.1–5.3).

- Cross-setting evaluation. The paper runs both instruction tuning and pretraining experiments and explicitly studies in‑domain vs out‑of‑domain generalization (e.g., MMLU‑val → MMLU‑test vs BBH, Fig. 1 & 2; Table 1), which aligns with the paper’s motivation about brittle offline curation

- Interpretable diagnostics. The distributional views (Fig. 3a–b on p.8) help illustrate how weighting changes the effective mixture and similarity landscape across epochs, consistent with an implicit curriculum narrative.

**Weaknesses:**

1. Compute accounting vs “negligible overhead.” The main text states ADAPT “incurs nearly zero additional overhead” (Sec. 1–2), but Appendix C quantifies ADAPT as 2k × 8N × P × E vs training’s typical 6N per token—i.e., ~33% extra FLOPs for ADAPT relative to a 6N baseline, which is not negligible. This discrepancy should be reconciled with a precise breakdown of where the +2N arises (similarity scoring, extra passes, etc.), and whether the same penalties are charged to other baselines (e.g., BM25 index building).

2. Use of evaluation data as anchors / potential leakage. For pretraining, ADAPT’s anchor/validation pool is built by sampling from the same downstream evaluation benchmarks (e.g., ARC‑C, COPA, MultiRC, etc.; Sec. 6.1, p.6–7). Even if only “validation” items are used, this design can bias models toward those tasks, weakening claims of broad generalization. A variant where anchors come from disjoint corpora (or different task families) is important to rule out target‑task leakage.

3. “100× fewer samples” and compute parity are under‑specified. The paper emphasizes using 100 unique examples but keeps training time nearly identical to R1 (15h37m vs 15h23m on Qwen3‑0.6B; §4) and shows only modest time savings versus AdaRFT (2–11%; Figure 13). It appears SPaRFT reduces unique training items but not the number of optimization steps. This weakens the “minimal resources” narrative and invites a stronger compute‑parity study (equal steps, equal tokens, and equal number of unique items for baselines).

4. On pretraining, ADAPT improves the average by 0.19–0.38 points over Uniform (Tables 2–3, pp.9), and performance is mixed on individual tasks. It’s not evident that these differences are robust: #seeds and statistical tests are not specified for Tables 2–3. For instruction tuning (Table 1, p.8), error bars are shown, but again the number of seeds and test methodology are unclear; several intervals overlap (e.g., ADAPT vs full SFT on BBH). Without significance analysis, claims of superiority are tentative.

5. Sec. 5 references “smooth gating,” normalization, and clipping but does not specify the exact functions, schedules, or hyperparameters (e.g., temperature, clipping thresholds, batch vs global normalization). Reproducibility would benefit from pseudocode and explicit details.

6. Baseline fairness and configuration. For DoReMi and RegMix, the paper states it uses domain weights from Lu et al. (2023) as selection ratios (Sec. 6.1). That sounds closer to using fixed weights than actually running the optimization procedures those methods prescribe, which could understate their performance relative to ADAPT. Please clarify and, ideally, run the baselines as originally intended. The LinUpper configuration and its cap parameter α could heavily impact performance; more detail and a parameter sweep would help substantiate the negative result (Table 2).

**Questions:**

1. Overhead accounting. Please reconcile the “nearly zero overhead” claim with ADAPT = 8N vs 6N FLOPs per token in Appendix C. What exact computations contribute to the +2N, and are equivalent costs charged to offline baselines (e.g., BM25 indexing, embedding passes)?

2. Anchors vs evaluation data. How much performance drops when anchors are drawn from unseen, disjoint distributions (e.g., natural‑language corpora not overlapping with evaluation tasks)? This is vital to support the generalization narrative.

3. Method details. What are the precise gating/normalization/clipping functions (and hyperparameters), and are weights normalized per‑batch or globally? Could you include pseudocode?

4. Baseline implementations. Did you run DoReMi/RegMix as originally proposed (with mixture‑weight optimization), or did you only apply fixed weights from another paper? If the latter, can you report results from the full algorithms?

5. Ablations. Can you provide ablations on (i) representation pooling (Eq. 9) vs mean pooling, (ii) anchor‑set size/composition, and (iii) choice of similarity metric (BM25 vs embedding vs model‑state cosine)?

---

> ### Author Response · Authors · 2025-11-27
>
> We are very grateful that you recognized our contribution and highlighted that “this framing is valuable for the community, where ‘hidden’ preprocessing costs are often ignored.” We also sincerely appreciate your thorough and constructive feedback. We address all of your concerns below and have incorporated the requested clarifications and ablations in the revised manuscript.
>
> ## W1 & Q1
> > Compute accounting vs “negligible overhead.” The main text states ADAPT “incurs nearly zero additional overhead” (Sec. 1–2), but Appendix C quantifies ADAPT as 2k × 8N × P × E vs training’s typical 6N per token—i.e., ~33% extra FLOPs for ADAPT relative to a 6N baseline, which is not negligible. This discrepancy should be reconciled with a precise breakdown of where the +2N arises (similarity scoring, extra passes, etc.), and whether the same penalties are charged to other baselines (e.g., BM25 index building).
>
> > Overhead accounting. Please reconcile the “nearly zero overhead” claim with ADAPT = 8N vs 6N FLOPs per token in Appendix C. What exact computations contribute to the +2N, and are equivalent costs charged to offline baselines (e.g., BM25 indexing, embedding passes)?
>
> We acknowledge the potential confusion caused by describing ADAPT's cost as $8N$ FLOPs. We have revised the manuscript to use a more accurate description of the cost:
>
> **Breakdown of the FLOPs Overhead**
>
> The $6N$ FLOPs represents the standard cost of model training (forward and backward passes). The theoretical overhead comes primarily from the **Quality Signal Computation**. For representation-based methods, this requires computing the **Training Sample Embedding** and the **Validation Set Embedding**.
>
> **Minimal Overhead in Practice**
>
> While a theoretical $2N$ overhead exists, we clarify that the practical overhead is minimal due to two critical amortization strategies:
>
> 1. The embedding for the training sample is obtained during the model's standard forward pass for training. It does not require an extra, separate forward pass. Therefore, this cost is amortized within the $6N$ training cost and incurs no additional time overhead for the forward computation.
>
> 2. The validation set embeddings do not need to be updated every single step. Since the model changes gradually, we perform this update periodically (e.g., once every $R$ steps or once per epoch). This amortizes the validation embedding generation cost over $R$ steps, making the average overhead per step negligible/minimal.
>
> Therefore, the actual additional computation overhead in ADAPT is minimal, arising only from the periodic updates, which is significantly less than a constant $33\%$ increase per step suggested by the $8N$ figure.
>
> To empirically validate that the computational cost of generating validation embeddings can be effectively amortized without performance loss, we conducted an ablation study comparing refresh intervals of $R=10$ and $R=40$. The results are summarized below:
>
> | R | MMLU(val) – MMLU(test) | MMLU(val) – BBH(test) |
> | :--- | :--- | :--- |
> | **10** | 50.2 ± 0.7 | 40.7 ± 1.6 |
> | **40** | 50.2 ± 0.5 | 40.7 ± 1.4 |
>
> As shown in the table, increasing the sparsity of updates from every 10 steps to every 40 steps results in identical average performance with comparable stability (standard deviations remain low). This confirms our hypothesis that model representations evolve gradually during training. Consequently, the computational cost of updating validation embeddings can be safely amortized over longer intervals (e.g., $R=40$ or higher), ensuring that the per-step FLOPs overhead of ADAPT remains negligible in practice.
>
> **Costs for Baselines and Other Quality Signals**
>
> Baselines: It is also important to note that offline baselines incur equivalent or greater computational costs, which are simply shifted to a preprocessing phase. Methods relying on offline embedding similarity require a separate preprocessing step to generate and index embeddings for all training samples. This is a mandatory computational requirement, external to the $6N$ training loop.
>
> ADAPT-BM25: As shown in our ablation study, the ADAPT-BM25 version (using BM25 similarity) does not incur the $2N$ overhead during the training loop. Its only overhead is the minimal, one-time cost of building the BM25 index.

---

> > ### Author Response · Authors · 2025-11-27
> >
> > ## W2 & Q2
> >
> > > Use of evaluation data as anchors / potential leakage. For pretraining, ADAPT’s anchor/validation pool is built by sampling from the same downstream evaluation benchmarks (e.g., ARC‑C, COPA, MultiRC, etc.; Sec. 6.1, p.6–7). Even if only “validation” items are used, this design can bias models toward those tasks, weakening claims of broad generalization. A variant where anchors come from disjoint corpora (or different task families) is important to rule out target‑task leakage.
> >
> > > Anchors vs evaluation data. How much performance drops when anchors are drawn from unseen, disjoint distributions (e.g., natural‑language corpora not overlapping with evaluation tasks)? This is vital to support the generalization narrative.
> >
> > As we mentioned in Line 338-340: “To construct this set, we sample 50 validation examples from each of eight evaluation benchmarks: ARC-C, COPA, Lambada, MultiRC, PiQA, RACE, SciQ, and Social IQA.” which means in our 15 benchmarks, the other 7 of them are disjoint corpora and hasnt' been seen in the anchor/validation pool. We believe the results on those benchmark can address your concern. We have updated Table 2 and Table 3, highlighting the results on the unseen tasks (the task not included in the validation set). We extract those results as follows:
> >
> > (Table 2 in the paper: 50B training token budget)
> >
> > | Tasks        | Uniform         | LinUpper        | DoReMi           | RegMix             | ADAPT            |
> > |--------------|------------------|------------------|-------------------|-------------------|-------------------|
> > | **Average (All) (↑)**    | 37.81 | 37.03 | 37.32 | 37.97 |  **38.00** |
> > | **Average (Unseen) (↑)** | 31.98 | 30.56 | 31.95 | 32.46 | **33.73** |
> >
> > (Table 3 in the paper: 100B training token budget)
> >
> > | Tasks        | Uniform | ADAPT  |
> > |--------------|---------|--------|
> > | **Average (All) (↑)**    | 37.98 | **38.36** |
> > | **Average (Unseen) (↑)** | 32.51 | **32.67** |
> >
> >
> >
> > The results in both tables show that the average performance on disjoint corpus of ADAPT outperforms other baselines, which validate our claim that generalisation of our online method ourperforms baselines.
> >
> >
> > ## W3
> >
> > > “100× fewer samples” and compute parity are under‑specified. The paper emphasizes using 100 unique examples but keeps training time nearly identical to R1 (15h37m vs 15h23m on Qwen3‑0.6B; §4) and shows only modest time savings versus AdaRFT (2–11%; Figure 13). It appears SPaRFT reduces unique training items but not the number of optimization steps. This weakens the “minimal resources” narrative and invites a stronger compute‑parity study (equal steps, equal tokens, and equal number of unique items for baselines).
> >
> > We believe this paragraph of comments was inadvertently copied from the review of a different submission, as the referenced details are entirely inconsistent with our manuscript: the methods (SPaRFT, AdaRFT), model (Qwen3-0.6B), and figure numbers do not appear anywhere in our paper. Our submission does not make a “100× fewer samples” claim, nor does it report the referenced training times or figures.
> >
> > We suspect this entire paragraph is unrelated to our work. If the reviewer has any questions regarding the claims specifically presented in our paper, we would be happy to provide the necessary clarification.

---

> > > ### Author Response · Authors · 2025-11-27
> > >
> > > ## W4
> > >
> > > > On pretraining, ADAPT improves the average by 0.19–0.38 points over Uniform (Tables 2–3, pp.9), and performance is mixed on individual tasks. It’s not evident that these differences are robust: #seeds and statistical tests are not specified for Tables 2–3.  For instruction tuning (Table 1, p.8), error bars are shown, but again the number of seeds and test methodology are unclear; several intervals overlap (e.g., ADAPT vs full SFT on BBH). Without significance analysis, claims of superiority are tentative.
> > >
> > >
> > > **Experimental details:**
> > >
> > > We apologize for not stating this explicitly. In the updated manuscript we now specify: Tables 1–3 are based on 3 independent runs per method. In pretraining, we use seeds 0, 1, and 2. In fine-tuning, we use seeds 3, 6, and 9. For each run, we retrain the model and re-run evaluation. We will also release our code and Weights & Biases logs after acceptance to further support reproducibility.
> > >
> > >
> > > **Error bars and significance analysis:**
> > >
> > > In Table 1 (fine-tuning), error bars denote the standard deviation across the 3 runs. In the revised Table 2 (pretraining), we also now report standard deviation for each method across the 3 runs.
> > >
> > > | Tasks                    | Uniform    | LinUpper   | DoReMi     | RegMix     |  ADAPT          |
> > > | ------------------------ | ---------- | ---------- | ---------- | ---------- | -------------- |
> > > | **Average (All) (↑)**    | 37.81±0.13 | 37.03±0.12 | 37.32±0.15 | 37.97±0.02 | **38.00±0.22**     |
> > > | **Average (Unseen) (↑)** | 31.98±0.09 | 30.56±0.28 | 31.95±0.15 | 32.46±0.39 | **33.73±0.39** |
> > >
> > > In both Tables 1 and 2, the gains on average performance of ADAPT over Uniform and other baselines exceed one standard deviation, supporting the robustness and significance of our improvements.
> > >
> > > **Performance–FLOPs trade-off:**
> > >
> > > We also now report FLOPs overhead for each method for pretraining. These numbers emphasize our central claim: online reweighting via ADAPT achieves the best performance–FLOPs trade-off, especially compared to multi-stage offline pipelines with heavy preprocessing and additional proxy training.
> > >
> > >
> > > | Tasks                    | Uniform    | LinUpper   | DoReMi     | RegMix     |  ADAPT          |
> > > | ------------------------ | ---------- | ---------- | ---------- | ---------- | -------------- |
> > > | **Average (All) (↑)**    | 37.81±0.13 | 37.03±0.12 | 37.32±0.15 | 37.97±0.02 | **38.00±0.22**     |
> > > | **Average (Unseen) (↑)** | 31.98±0.09 | 30.56±0.28 | 31.95±0.15 | 32.46±0.39 | **33.73±0.39** |
> > > | **FLOPs overhead (↓)**   | 0                | 0                | 4.92×10¹⁹        | 3.072×10¹⁸       | ≪1.1×10¹⁵         |
> > >
> > >
> > > ## W5 & Q3
> > >
> > > > Sec. 5 references “smooth gating,” normalization, and clipping but does not specify the exact functions, schedules, or hyperparameters (e.g., temperature, clipping thresholds, batch vs global normalization). Reproducibility would benefit from pseudocode and explicit details.
> > >
> > > > Method details. What are the precise gating/normalization/clipping functions (and hyperparameters), and are weights normalized per‑batch or globally? Could you include pseudocode?
> > >
> > > We appreciate the chance to clarify this section and provide further details. We have added pseudocode to the paper for your reference and have also expanded the method details in the revised manuscript to improve clarity.
> > >
> > > **Embedding Normalization**: We apply L2 normalization to embeddings $\phi(x)$ and $\phi(v)$ before computing cosine similarities to ensure scale-invariant similarity measurements. Specifically, for each embedding vector, we compute $\phi(x) \leftarrow \phi(x) / \max(\|\phi(x)\|_2, \epsilon)$ where $\epsilon$ prevents division by zero. This normalization step is distinct from weight normalization and serves to standardize the embedding space for similarity computation.
> > >
> > > **Smooth Gating Function:** We use a sigmoid function with temperature scaling:
> > >
> > > $$w_t(i) = \sigma\left(\frac{s_{ADAPT}(x_i)}{\tau}\right) = \frac{1}{1 + \exp(-s_{ADAPT}(x_i) / \tau)}$$
> > >
> > > where $\sigma(\cdot)$ is the sigmoid, $\tau = 0.1$ (default, minimum $10^{-6}$ for numerical stability), and $\epsilon = 10^{-8}$ is used in L2 normalization and pooling operations.
> > >
> > > The sigmoid transformation maps similarity scores to absolute weights in the interval $[0, 1]$ without requiring normalization across samples in the batch. This design choice ensures that the weight assigned to each sample $x_i$ depends solely on its similarity score $s_{ADAPT}(x_i)$ relative to the anchor set, rather than its rank within the current mini-batch. Consequently, a sample with a given similarity score receives the same weight regardless of whether it appears in a high-quality or low-quality batch, making the weighting mechanism robust to batch-level variations in data quality. This contrasts with normalized weighting schemes (e.g., softmax normalization) where weights are relative to other samples in the same batch, potentially amplifying or dampening effects based on batch composition.

---

> > > > ### Author Response · Authors · 2025-11-27
> > > >
> > > > ## W6
> > > > > Baseline fairness and configuration. For DoReMi and RegMix, the paper states it uses domain weights from Lu et al. (2023) as selection ratios (Sec. 6.1). That sounds closer to using fixed weights than actually running the optimization procedures those methods prescribe, which could understate their performance relative to ADAPT. Please clarify and, ideally, run the baselines as originally intended.
> > > >
> > > > We confirm that our experimental setup was designed to ensure a controlled, fair, and maximally faithful comparison against all baselines.
> > > >
> > > > ----
> > > > **DoReMi and RegMix (Offline Domain-Based Methods)**
> > > >
> > > > We emphasize that our entire experimental setting (data, model size, pre-training parameters, etc.) is identical to the configuration used in the previous work we refer to. The weights we adopted were not arbitrarily fixed, but were obtained through **fully executing the original optimization procedures under this same configuration**:
> > > > DoReMi weights were obtained by training proxy models for 10k steps. RegMix weights were obtained by training 200 1M proxy models using CC as the target domain.
> > > >
> > > > DoReMi and RegMix are **offline, multi-stage methods**, where the optimization process yields a set of **fixed domain weights to guide subsequent training**. Given the complete consistency of our experimental configuration, adopting their derived weights is the most robust strategy. This approach ensures we **avoid the potential randomness and variance** that might arise from reproducing their complex.
> > > >
> > > > As we mentioned in Lines 340-342, using these pre-calculated weights as the sampling ratios for our final model training is the most **faithful** way to reproduce their effect. **If the baseline algorithm is robust, running it on the same data should give consistent results.**
> > > >
> > > > All reported results for these baselines are derived from our own, independent training runs (with multiple seeds) using these fixed domain weights, ensuring the conclusions are robust.
> > > >
> > > > -----
> > > > > The LinUpper configuration and its cap parameter α could heavily impact performance; more detail and a parameter sweep would help substantiate the negative result (Table 2).
> > > >
> > > > **LinUpper (Online Sample-Based Method)**
> > > >
> > > > LinUpper is an online, sample-based method, and we rigorously followed its original implementation. We adopted the **exact same $\alpha$ value used in the original LinUpper paper** to ensure our configuration is **consistent and comparable to the authors' own reported optimal settings**.
> > > >
> > > > We hypothesize that LinUpper’s negative performance may come from its within-batch normalization: if an entire batch is high-quality (or low-quality), the normalized weights become similar to another batch with a different absolute quality level, which can blunt the effect of absolute quality signals.
> > > >
> > > > ---
> > > > Overall, we have been careful to configure all baselines in a way that is faithful to the original methods and to report our own training runs (with multiple seeds) for each, ensuring both fair comparison and robust conclusions.

---

> ### Author Response · Authors · 2025-11-27
>
> ## Q4
>
> > Ablations. Can you provide ablations on (i) representation pooling (Eq. 9) vs mean pooling, (ii) anchor‑set size/composition, and (iii) choice of similarity metric (BM25 vs embedding vs model‑state cosine)?
>
> We thank the reviewer for suggesting these valuable ablation studies. We have conducted the requested experiments and incorporated the detailed results into the revised manuscript. The table below summarizes the key ablation results in comparison to the baselines and our proposed ADAPT method:
>
> | Method     | Average (↑)     |
> |------------|-----------------|
> | Uniform    | 37.81±0.13      |
> | LinUpper   | 37.03±0.12      |
> | DoReMi     | 37.32±0.15      |
> | RegMix     | 37.97±0.02      |
> | ADAPT      | 38.00±0.22      |
> | **ADAPT-MP** (mean pooling)   | **37.80±0.41**      |
> | **ADAPT-SV** (smaller validation set)   | **37.98±0.34**      |
> | **ADAPT-SP** (SlimPajama validation anchors)  | **37.97±0.27**      |
> | **ADAPT-BM25**   | **38.05±0.21**      |
>
> (i) Replacing our weighted mean-pooling with a uniform mean-pooling (denoted as ADAPT-MP) led to a performance decrease from $38.00 \pm 0.22$ to $\mathbf{37.80 \pm 0.41}$. This demonstrates the necessity and advantage of our weighted mean-pooling approach. The learned weights effectively capture the importance of representations at different training steps, leading to superior overall performance compared to unweighted averaging.
>
> (ii) We performed ablations on both the size and composition of the anchor set to demonstrate the robustness of our method.
>
> *Size:* We significantly reduced the anchor set size, sampling only 20 validation examples from each of the eight evaluation benchmarks, totaling 160 samples. The performance (shown as ADAPT-SV, smaller validation) was $\mathbf{37.98 \pm 0.34}$. The performance fluctuated only slightly (a decrease of 0.02) and still significantly outperformed other baselines. This confirms the high stability of our proposed method with respect to the anchor-set size.
>
>
> *Composition:* We changed our anchor set to the validation set from SlimPajama (a general, unrelated corpus validation set). The performance (shown as ADAPT-SP, SlimPajama) was $\mathbf{37.97 \pm 0.27}$. The performance again fluctuated only slightly (a decrease of 0.03), maintaining a strong result. This demonstrates the generalization and robustness of ADAPT, as it performs well even when the anchor set is sourced from a general-purpose corpus validation set rather than task-specific validation data.
>
>
> (iii) We replaced the model-state embedding with the BM25 score as the quality signal within the ADAPT framework. The result (shown as ADAPT-BM25) was $\mathbf{38.05 \pm 0.21}$. Surprisingly, using BM25 as the quality signal not only outperformed all other baselines but also slightly surpassed our original model-state embedding approach ($38.05$ vs $38.00$). This highlights the flexibility and efficacy of the ADAPT framework, showing that it can successfully leverage diverse quality signals (both dense embeddings and sparse lexical metrics) to effectively modulate training. We have updated Table 2 to include detailed per-task results and FLOPs comparison.
>
>
> ------
>
> Once again, we thank the reviewer for the very helpful comments and suggestions. They have improved the clarity, completeness, and robustness of our paper. We sincerely hope that the above responses sufficiently address all your concerns. Should there be any inaccuracy in our understanding, or if you have any further questions on any detail, we would be very happy to provide additional clarification and discussion.

---

### Author Response · Authors · 2025-12-04
**General Response**

We thank the Area Chair and the reviewers for their time and constructive feedback. We are encouraged that the reviewers found our unified framework **“valuable for the community” (Reviewer SsJ4)**, the problem **“timely” (Reviewer osmP)**, and the experimentation **“extensive and thorough” (Reviewer osmP)**.

We have updated the manuscript to address all raised concerns. Below, we summarize our key improvements and responses, specifically highlighting how we addressed the questions regarding computational overhead, statistical significance, and comparison with baselines.

 1. **Generalisation, Robustness and Statistical Significance (SsJ4 & P23q)**

We have updated **Table 1 (Instruction Tuning)** and **Table 2 (Pretraining)** to report results averaged over **3 independent random seeds** with standard deviations. ADAPT consistently outperforms baselines beyond the margin of error. For pretraining, ADAPT outperforms RegMix ($38.00$ vs $37.97$) and LinUpper ($38.00$ vs $37.03$). Especially on unseen tasks (tasks not used in the anchor set), ADAPT shows a decisive advantage ($33.73$ vs $32.46$ for RegMix), proving that our method does not just memorize the validation set distribution but learns robust features.

2. **Extended Ablations and Additional Experiments on Scalability (SsJ4 and P23q)**

We added comprehensive ablations to demonstrate the robustness of our design choices:

* **Pooling Strategy:** We showed that our weighted mean-pooling outperforms standard uniform pooling ($38.00$ vs $37.80$).
* **Anchor Set Size:** We demonstrated that ADAPT remains robust even when the anchor set size is significantly reduced (160 samples) or when using a generic corpus validation set (SlimPajama validation) instead of task-specific anchors.
* **Refresh Intervals:** We demonstrated that the per-step FLOPs overhead of ADAPT is minimal in practice.

In addition, we conducted new experiments on Llama-2-13B. ADAPT significantly outperforms the baseline (55.4% vs 52.5% on MMLU), proving our method scales effectively to larger architectures.

3. **Qualitative Analysis (osmP)**

We expanded Appendix to show that ADAPT selects semantically high-quality data regardless of format, whereas offline methods often overfit to the specific format of validation examples.

4. **Details of Method and Implementation (SsJ4)**

We added **Pseudocode** in the Appendix, and provided explicit hyperparameter details in our rebuttal.

5. **Respectful Clarification (P23q)**

We respectfully pointed out that we *did* compare against a relevant online method (LinUpper) and outperformed it significantly. We clarified that the other suggested papers (self-improvement/synthetic data) operate in fundamentally different settings than raw corpora curation. We validated on both pretraining (120M & 1.1B models) and instruction tuning (Llama-2-7B and 13B), covering the full LLM lifecycle.

6. **Clarification on Pareto Frontier (SsJ4)**

We have demonstrated that ADAPT establishes a superior **Pareto Frontier** across *both* training paradigms by structurally minimizing overhead:
* Zero Overhead for Training Embeddings: A critical design efficiency is that **the embedding for the training sample is obtained during the model's standard forward pass for training.** It does not require an extra, separate forward pass. Therefore, this cost is amortized within the training cost and incurs no additional time overhead for the forward computation.
* Instruction Tuning: As shown in Figure 2, ADAPT consistently achieves higher accuracy per FLOP. Offline methods like LESS incur significant computational costs due to the requirement of calculating gradients over the entire data pool for scoring. Furthermore, while heuristics like BM25 suffer from a "ceiling effect" and saturate quickly, ADAPT continues to improve model performance as training progresses, offering a superior trade-off for high-performance training.
* Pretraining: Offline baselines like DoReMi and RegMix require massive pre-computation (up to $4.92 \times 10^{19}$ FLOPs) to train proxy models. In contrast, ADAPT eliminates this pre-processing stage entirely. By leveraging the amortized cost mechanism described above, ADAPT achieves negligible total overhead ($\ll 1.1 \times 10^{15}$ FLOPs) while outperforming baselines in accuracy.

### **Summary of Revisions**

In the updated manuscript, we have:
* **Statistical Rigor**: Extended the multi-seed evaluation protocol from instruction tuning to pretraining.
* **Included FLOPs Breakdown:** Explicitly listed FLOPs overhead for all methods to highlight ADAPT's efficiency.
* **Expanded Ablations:** Added results for Pooling strategies, Anchor Set variations, and Metric choices (BM25 vs Embedding).
* **Added Pseudocode:** Included in Appendix for reproducibility.
* **Extended Experiments:** Added results for different sizes of models.

---

### Meta-Review · Area_Chair_6ZuX · 2026-01-05

**Summary:**

The authors propose a method that dynamically re-weighs individual samples during training. The authors claim that they achieve better per FLOP performance compared to baselines, other online reweighing strategies, and also achieve better out of domain generalization.

Many concerns were addressed with clarifications, and additional experiments. Overall, the paper is quite strong. It addresses the issues with offline reweighing methods, and improves upon the computational complexity of the online ones. The proposed method has good OOD performance.

Key remaining weaknesses:
 - Reviewers noted that performance gains are really minimal for in-distribution evaluation. The authors' strongest argument is efficiency and robustness, not performance on seen tasks.
 - Implementing an online, per-sample reweighing loop with dynamic validation embedding updates is significantly more complex to engineer than simple offline filtering. While the authors commented on the computational overhead and argued that it is minimal, the engineering aspect was not really addressed or commented much on.
 - conceptual novelty is incremental (effectively a better implementation of reweighing).

While some (like the first one) weaknesses cannot really be addressed, I would encourage the authors to comment on the other two, and not overstate their contributions (especially regarding the "online" aspect).

**Reviewer Concerns:**

Reviewer SsJ4 questioned the claim of "nearly zero overhead" of the proposed method. The authors provided an explanation of amortization, and empirical evidence supporting sparse updates. The reviewer also asked about the anchor set formation and whether any data leakage occurs (the authors clarified why it doesn't). I would say the concerns were partially addressed.

Reviewers P23q and SsJ4 noted that the performance gains in pretraining were small. The authors presented new results from 3 runs from different seeds, strengthening the results and showing consistent improvement. However, the improvement is still very small.

Reviewer P23q noted the paper claimed "online" as a novel contribution, ignoring existing online reweighing algorithms. The authors pointed to already existing comparison, though the claims should still be adjusted properly in the paper. So I would say this one is partially addressed.

Reviewer's osmP concern was addressed with qualitative examples that provided new insights into why the method works better for OOD tasks. They also noted that a simple heuristic, BM25, beat most baselines at a very low computational cost.

The authors also added experiments using Llama-2-13B to respond to the request for larger models.

**Reviewer Scores:**

Reviewer P23q is expected to increase their score, since their concerns about limited improvement and missing baselines were addressed by the new data on unseen tasks. Their concern about overstating contributions can be  (and should be ) addressed with minor reframing.

Reviewer SsJ4 is expected to increase their score, all reasonable concerns were addressed.

Reviewer osmP is expected to maintain or increase their score, as their key concerns (performance vs flops trade-off, qualitative analysis, clarification on definitions) seemed to have been mostly addressed.

---

### Decision · Program_Chairs · 2026-01-26

Accept (Poster)